**DOI: 10.1038/ncomms13954**　　**OPEN**

# Marine origin of retroviruses in the early Palaeozoic Era

Pakorn Aiewsakun[1] & Aris Katzourakis[1]

Very little is known about the ancient origin of retroviruses, but owing to the discovery of their ancient endogenous viral counterparts, their early history is beginning to unfold. Here we report 36 lineages of basal amphibian and fish foamy-like endogenous retroviruses (FLERVs). Phylogenetic analyses reveal that ray-finned fish FLERVs exhibit an overall co-speciation pattern with their hosts, while amphibian FLERVs might not. We also observe several possible ancient viral cross-class transmissions, involving lobe-finned fish, shark and frog FLERVs. Sequence examination and analyses reveal two major lineages of ray-finned fish FLERVs, one of which had gained two novel accessory genes within their extraordinarily large genomes. Our phylogenetic analyses suggest that this major retroviral lineage, and therefore retroviruses as a whole, have an ancient marine origin and originated together with, if not before, their jawed vertebrate hosts >450 million years ago in the Ordovician period, early Palaeozoic Era.

[1] Department of Zoology, University of Oxford, Oxford OX1 3PS, UK. Correspondence and requests for materials should be addressed to A.K. (email: aris.katzourakis@zoo.ox.ac.uk).

Retroviruses are a group of medically and economically important viruses (family *Retroviridae*) that infect a wide range of animals from fish to humans[1,2], and can occasionally leave genomic fossils within their host genome, known as endogenous retroviruses (ERVs). ERVs are relics of past infections, resulting from viral genomic integrations that occur in host germ-line cells, and are in turn passed down from parents to offspring as part of the host genome[3]. Owing to the increasing availability of animal whole genomes, more and more ancient ERVs have been discovered, allowing the early history of retroviruses to be examined like never before[3]. The oldest age estimates directly inferred for retroviruses are $\sim 100$ million years (Myr) old, derived from analyses of mammalian retroviruses and ERVs[4–6]. Beyond this point in time, the origins of retroviruses remain unclear due to the difficulties of accurately identifying ancient retroviral integrations, and the limitations of extrapolating from extant retroviral sequence data.

Retroviruses are extremely widespread among vertebrates[1,2], raising the possibility that they might be as old as their vertebrate hosts. However, retroviruses frequently cross species even across vertebrate classes, as evidenced by phylogenetic analyses incorporating distantly related ERVs[1,2]. Thus, retroviruses could be $\sim 100$ Myr old and have been transmitted across vertebrates, or may date back to the origins of vertebrates at $\sim 460–550$ Myr ago (Ma)[7–9]. To better examine and estimate the date of origin of retroviruses necessitates analyses of retroviruses whose evolutionary dynamics are well understood, and because of this, we turn to foamy viruses (FVs).

FVs are a unique subgroup of retroviruses (genus *Spumaretrovirus*) that are characterized by an extremely stable history of co-speciation with their mammalian hosts, at least since the origin of eutherians $\sim 100$ Myr ago[5,6]. This unique evolutionary feature, together with the high availability of FV molecular data, allows their evolutionary dynamics to be described in unprecedented detail, making FVs one of the most important models of retroviral macroevolution. Similar to other retroviruses, FVs occasionally leave viral genomic fossils in their host genomes. Three mammalian endogenous FVs have been discovered to date, including the prosimian aye-aye endogenous FV (*Daubentonia madagascariensis*; PSFVaye[6,10]), sloth endogenous FV (*Choloepus hoffmanni*; SloEFV[5]) and Cape golden mole endogenous FV (*Chrysochloris asiatica*; ChrEFV[6,11]). On the basis of protein sequence similarity, a number of ERVs have also been identified as FV-like in fish genomes, including coelacanth (*Latimeria chalumnae*; CoeEFV[12]), platyfish (*Xiphophorus maculatus*; platyfishEFV[13]) and zebrafish (*Danio rerio*; DrFV-1 (ref. 14)). Phylogenetic analyses showed that CoeEFV is positioned basal to mammalian FVs[12], and that DrFV-1 and platyfishEFV form a monophyletic clade, basal to CoeEFV and mammalian FVs[13]. This phylogenetic pattern matches perfectly that of vertebrates, raising the possibility of an ancient marine origin of this major retroviral lineage[12,13]. However, unlike in the case of mammalian FVs, this inferred large-scale co-speciation history could be misleading, as limited data sampling may have obscured a history of cross-species transmissions. Outside of mammals, there is no statistical evidence of congruence of host and FV phylogenetic topologies, and therefore the two scenarios cannot be distinguished. Additional ERV data from multiple vertebrate classes could help resolve the ancient origins of FVs, and also shed light on the early history of retroviruses as a whole.

Here we present 36 novel lineages of amphibian and fish FV-like ERVs (FLERVs), some of which have the largest known retroviral genomes. We also re-examine the co-evolutionary history of this major retroviral lineage with their vertebrate hosts. To reconstruct the evolutionary history of these viruses, we incorporate a recently developed approach that can account for the fact that the rates of viral evolution are time-dependent[15–18], appearing to decay over time following a power-law pattern[16]. By overcoming the limitations of extrapolating across different timescales, incorporating multiple endogenous FVs, and leveraging the stable evolutionary dynamics of FVs, we show that this major group of retroviruses emerged $>450$ million years ago in the early Palaeozoic Era, coinciding with the origin of jawed vertebrates. To our knowledge, this is the first temporal evidence indicating that retroviruses are at least as old as their jawed vertebrate hosts.

## Results

**Discovery of FLERVs in amphibian and fish genomes.** By using tBLASTn and the reverse transcriptase (RT) protein of CoeEFV as a screening probe, 1,752 RT sequences were retrieved from publically accessible nucleotide GenBank databases, including the database of GenBank non-redundant nucleotide sequences, expressed sequence tags, high throughput genomic sequences, whole genome shotgun sequences and transcriptome shotgun assembly sequences. Together with 304 additional RT sequences from seven retroviral genera, phylogenetic analyses showed that 161 sequences from 28 distinct species (4 salamanders, 1 frog, 20 ray-finned fish, 2 lobe-finned fish and 1 shark) cluster together with the known mammalian FVs and fish ERVs previously identified as FV-like. We therefore only focused on these 161 retroviruses, and defined them as FLERVs.

To identify potentially full-length FLERVs, we extended the sequences on both sides and BLASTed the sequences against themselves to search for long terminal repeats (LTRs)—the key characteristic feature of ERVs that defines the boundary of the element. By doing so, we discovered one potentially full-length novel FLERV in the eastern newt (*Notophthalmus viridescens*) and four in the midas cichlid (*Amphilophus citrinellus*) genomes. We designated them 'NviFLERV' and 'AciFLERV', respectively. Furthermore, upon the ERV insertion, the target site DNA is also duplicated, resulting in small target site duplications (TSDs) flanking the ERVs[19]. We were able to identify NviFERLV's TSDs as well as those of the four AciFLERVs (Supplementary Note 1), supporting that they are *bona fide* ERVs, and are not recombinants of multiple elements. Furthermore, we also found five potentially full-length FLERVs in the zebrafish (*Danio rerio*) and three in the West Indian Ocean coelacanth (*Latimeria chalumnae*) genomes. These ERVs have already been described as DrFV-1 (ref. 14) and CoeEFV[12], respectively.

To search for additional FLERVs, we screened the five publically accessible nucleotide databases again by using the RT protein sequences of SloEFV, NviFLERV and AciFLERV. Thirty-two RT sequences from four additional ray-finned fish species were identified as FV-like, phylogenetically grouping with mammalian FVs and fish FLERVs. By expanding the sequences and based on the identification of their LTRs and TSDs, we discovered two more potentially full-length FLERVs in annual killifish (*Austrofundulus limnaeus*), and designated them 'AliFLERV'. We were able to identify TSDs of only one AliFLERV however, as the contig that harbours the other AliFLERV does not contain complete LTRs (Supplementary Note 1). In total, 193 sequences from 32 unique vertebrate species were identified as FLERVs (Supplementary Table 1), and their phylogeny together with other retroviruses is shown in Fig. 1. Our analyses show that they are indeed most closely related to the known FVs and FLERVs with strong support (bootstrap clade support = 95% and Bayesian posterior probability = 1.00). Detailed phylogenetic relationships among FVs and FLERVs, inferred from RT sequences are shown in Supplementary Fig. 1.

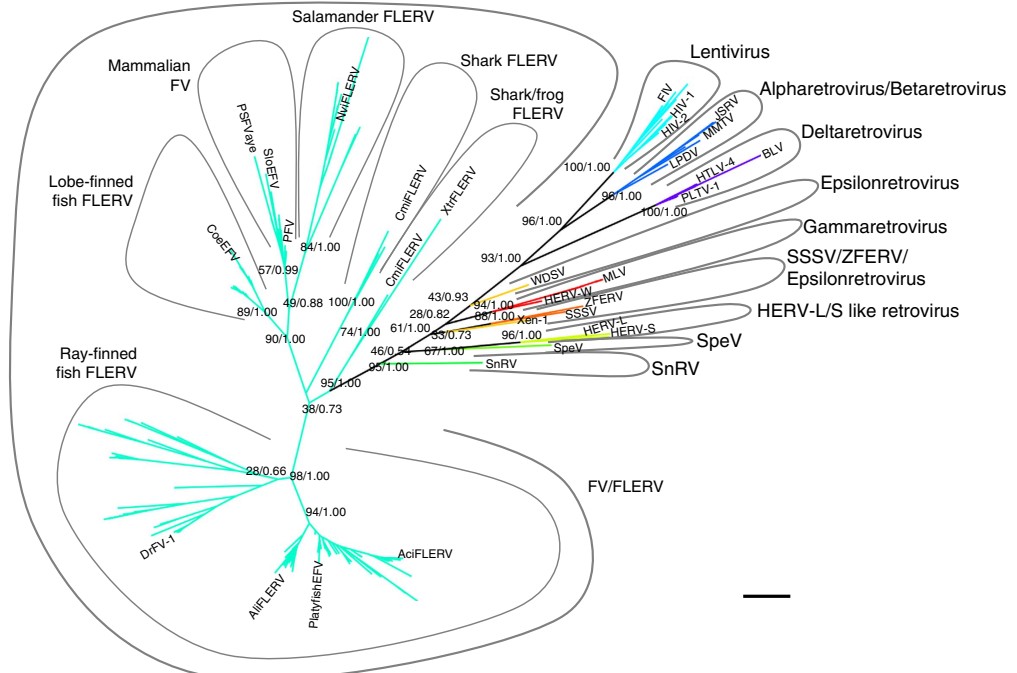

**Figure 1 | Retroviral phylogeny illustrating how FVs and FLERVs relate to other retroviruses.** The un-rooted phylogenies were estimated from a reverse transcriptase protein alignment. Since both Bayesian maximum clade credibility and maximum-likelihood trees have very similar topologies, only the Bayesian tree is shown. The scale bar (black solid line; bottom right corner) represents genetic divergence of length 0.2 in units of amino acid substitutions per site. The numbers on the nodes are bootstrap (first) and Bayesian posterior probability (second) clade support values.

Our RT phylogeny strongly supports monophyletic clades of salamander, lobe-finned fish and ray-finned fish FLERVs (bootstrap clade support >84% and Bayesian posterior probability = 1.00 for all clades). The clade of mammalian FVs was strongly supported by the Bayesian phylogenetic analysis (Bayesian posterior probability = 0.99), but not by the maximum-likelihood method (bootstrap clade support = 57%). On the other hand, we found that the frog FLERV clusters together with shark FLERVs (bootstrap clade support = 74% and Bayesian posterior probability = 1.00), and the shark FLERVs appear to form two separate clades; however, the latter phylogenetic pattern is not strongly supported (bootstrap clade support = 38% and Bayesian posterior probability = 0.73). We also observed that mammalian FVs, salamander FLERVs and lobe-finned fish FLERVs cluster together (bootstrap clade support = 90% and Bayesian posterior probability = 1.00), and ray-finned fish and shark FLERVs are basal to this clade, reflecting the host history. The phylogeny also shows that mammalian FVs are more closely related to salamander FLERVs than lobe-finned fish FLERVs, mirroring the host evolutionary relationship and thus consistent with a long-term co-speciation history between this retroviral lineage with their tetrapod hosts. However, again, the support for this relationship is low (bootstrap clade support = 49% and Bayesian posterior probability = 0.88), likely because the RT sequences used to reconstruct the tree were short (130 aa), and thus, the topology of this tree should not be over-interpreted. Indeed, as noted by others[5,20], RT trees are suitable only for broad classification of viruses. Below, we reconstructed a phylogeny of longer Pol protein sequences to determine the relationship among FVs and FLERVs more precisely (see 'Results' section: Phylogenetic analyses). Interestingly, our RT phylogenetic analysis shows that some fish and amphibian genomes harbour two FLERV lineages, including the genomes of Japanese fire belly newt (*Cynops pyrrhogaster*), fathead minnow (*Pimephales promelas)*, Atlantic cod (*Gadus morhua*), midas cichlid

(*Amphilophus citrinellus*) and Amazon molly (*Poecilia Formosa*), as well as West Indian Ocean coelacanth (*Latimeria chalumnae*) (Supplementary Fig. 1). To our knowledge, no known mammals have been shown to harbour two distinct lineages of FVs.

**Characterization of full-length FLERVs.** To characterize our novel amphibian and fish FLERVs, we first focused on the potentially full-length FLERVs, NviFLERV, AciFLERV and AliFLERV, and used mammalian FVs as the main reference for comparison (Fig. 2; see detailed genome annotation in Supplementary Fig. 2 and Supplementary Note 1). We did not characterize DrFV-1 and CoeEFV as they have been previously described[12,14]. We found that many of the potentially full-length NviFLERV, AciFLERV and AliFLERV contain large insertion/ deletion mutations, in-frame stop codon and frameshift mutations as well as transposable elements (TEs), making genome annotation difficult and potentially inaccurate. To address this problem, we used the full-length elements to retrieve additional (fragmented) elements from their respective genomes using BLASTn, and reconstructed the maximum-likelihood sequence of the basal node on the phylogeny. This inferred ancestral sequence was used for genome annotation. Unfortunately, only two NviFLERV elements were found in *N. viridescens*, designated 'NviFLERV-1' and 'NviFLERV-2', and thus its ancestral sequence could not be reconstructed reliably.

NviFLERV-1 is a complete full-length FLERV (9,200 nt), containing putative *gag*, *pol* and *env* genes that are similar to those of simian FVs (SFVs), flanked by 5′- and 3′-LTRs. Situated on the 3′ end of the 5′-LTR is a tRNA$^{Asn}$-utilising putative primer binding site (PBS), identified via sequence similarity, similar to that of prosimian galago FV[6], but different from those of other mammalian FVs, which have tRNA$^{Lys}$-utilising PBSs. Several in-frame stop codons and frameshift mutations were found in protein coding regions,

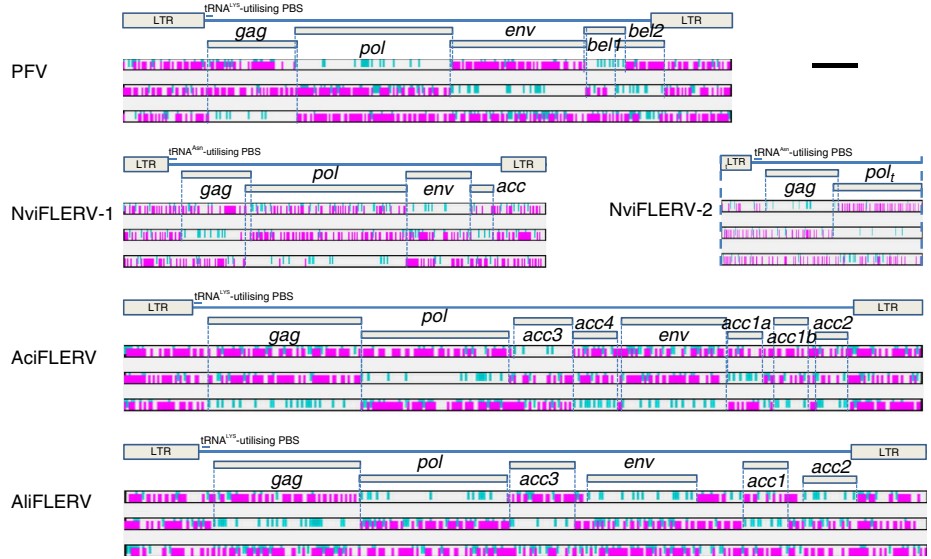

**Figure 2 | Genomic organizations of FVs and FLERVs.** NviFLERV-1 and NviFLERV-2 are endogenous viral elements present within the genome of *N. viridescens*. This is in contrast to AciFLERV and AliFLERV, which were annotated by using their maximum-likelihood ancestral sequences, reconstructed from 9 and 23 elements, respectively. Under each element are the distributions of stop codons (TAA, TAG and TGA; pink) and start codon (ATG; blue) in frame +1 (top), +2 (middle) and +3 (bottom), used to determine potential protein coding regions (blue thin vertical dashed lines). 't'-subscription indicates that the domain is truncated (5′-truncated: preceding the domain name; 3′-truncated: following the domain name; blue thick vertical dashed lines indicate where domains are truncated). Prototype foamy virus (PFV) was included as a reference. The scale bar (black solid line; top right corner) represents a nucleotide length of 1 kilobase.

indicating that it is defective, typical for an ERV. Between the *env* and 3′-LTR is a stretch of uniquely-mapping 492 nt, 164 aa, that does not exhibit similarity to any genes apart from a gene of Porcine reproductive and respiratory syndrome virus (BLASTx: AEQ61854; E = $6 \times 10^{-4}$). This relatively high E-value of $6 \times 10^{-4}$ makes it unlikely that they are homologues; nevertheless since mammalian FVs contain accessory genes in this region (*bel1* and *bel2*), and because it exhibits some similarity to a viral gene, we hypothesize that this nucleotide region might be an accessory gene of the progenitor of NviFLERV-1. Indeed, further analyses revealed that other salamander FLERVs also contain this gene (Table 1, see below), supporting that it has a viral origin and is not a host gene. We in turn annotated the gene as an 'acc' gene.

The fact that NviFLERV-1 has paired 5′- and 3′-LTRs enables us to estimate its age. Due to the process of retroviral genomic integration, the paired LTRs are at first identical, but after becoming endogenous, the two gradually diverge from one another[3]. Assuming that LTRs evolve neutrally after endogenisation, it is possible to calculate the integration date of the element, ie, its age, from the genetic distance of the two paired LTRs. The synonymous substitution rate of amphibian evolution was estimated to be ~0.924–1.53 × $10^{-9}$ substitutions per site per year[21]. Since synonymous substitutions are largely neutral, we considered it to be a reasonable approximation for a neutral rate of evolution of amphibians. We estimated the pairwise distance of the paired LTRs to be 4.4%. Given this estimated LTR pairwise distance and the synonymous substitution rate, we estimated NviFLERV-1 to be ~14 Myr old. The fact that we could identify the TSDs of NviFLERV-1 indicates that NviFLERV-1 is likely a *bona fide* ERV and its ancient age estimate is not an artefact of recombination between multiple ERVs.

NviFLERV-2 is a truncated element (4,391 nt), containing a 5′-truncated 5′ LTR, a complete *gag* gene, and a partial *pol* gene. A tRNA^Asn-utilising PBS similar to that of NviFLERV-1, and prosimian galago FV, was also identified by via sequence similarity. In contrast to NviFLERV-1, NviFLERV-2 *gag* and

*pol* genes do not contain in-frame stop codon or frameshift mutations. This raises two possibilities: either (i) that NviFLERV-2 is so young that it has not yet gained any in-frame stop codon or frameshift mutations or (ii) that it is a result of viral genomic contamination during the eastern newt genomic sequencing. To evaluate these possibilities, we screened short read archives for reads that span across the viral–host junction. Numerous reads mapping the junction were found, supporting that NviFLERV-2 is indeed a (very young) ERV, and not a result of viral genomic contamination.

Unlike NviFLERVs, we found 9 elements of AciFLERVs and 23 elements of AliFLERVs, allowing us to reconstruct their ancestral sequence for better genomic annotation. The original elements contain in-frame stop codon and frameshift mutations as well as TEs and large insertion/deletion mutations, confirming that they are indeed ERVs. The inferred ancestral sequence of AciFLERV and AliFLERV are 17,409 and 17,490 nt long, respectively, much longer than the length of typical mammalian FV genomes, ~10 kb. It is noteworthy that these lengths are not inflated by TEs and/or large insertions as they were removed during the reconstruction of ancestral sequences. By investigating the distribution of start and stop codons, we determined AciFLERV to have 8 open reading frames (ORFs; ORF-1 to -8, from the 5′ end of the genome) flanked by 5′- and 3′ LTRs. Again, this is very different from mammalian FVs, which has only 5 ORFs, including (from the 5′ end) *gag*, *pol* and *env* genes followed by two accessory genes *bel1* and *bel2*.

On the basis of sequence similarity and manual sequence inspection, we identified a tRNA^Lys-utilising PBS on the 3′ end of the 5′-LTR, similar to those utilized by most mammalian FVs, and determined that ORF-2, and ORF-5 are *pol* and *env* genes, respectively. We in turn annotated ORF-1 as *gag* gene, and ORF-6, −7 and −8 as accessory genes based on our knowledge of retroviral genomic structure. Further inspection revealed that ORF-6 and −7 proteins are highly similar (BLASTp: E = $1 \times 10^{-26}$), indicative of paralogy. We hence

**Table 1 | Genomic structure of amphibian and fish FLERVs.**

| Class | Species | gag | pol | acc3 | acc4 | env | Accessory gene (acc) acc1 | acc2 |
|---|---|---|---|---|---|---|---|---|
| Amphibia | Cynops pyrrhogaster | ○ | ○ | | | ○ | ○† | |
| | | ○ | ○ | | | ○ | ○† | |
| | Hynobius retardatus | ○ | ○ | | | ○ | | |
| | Lissotriton vulgaris | ○ | | | | | | |
| | Notophthalmus viridescens | ○ | ○ | | | ○ | ○† | |
| | Pleurodeles waltl | ○ | ○ | | | ○ | ○† | |
| | Xenopus tropicalis | | ○ | | | | | |
| Chondrichthyes | Callorhinchus milii | | ○ | | | | | |
| Osteichthyes | Amphilophus citrinellus | | ○ | | | | | |
| | | ○ | ○ | ○ | ○ | ○ | ○‡ | ○ |
| | Anoplopoma fimbria | ○ | ○ | ○ | | ○ | | ○ |
| | | | ○ | | | | | |
| | Austrofundulus limnaeus | ○ | ○ | ○ | | ○ | ○ | ○ |
| | Cynoglossus semilaevis | ○ | ○ | ○ | | ○ | ○ | |
| | Cyprinus carpio | | ○ | | | | | |
| | Dicentrarchus labrax | ○ | ○ | ○ | ○ | | ○ | ○ |
| | Esox lucius | | ○ | | | | | |
| | Fundulus heteroclitus | ○ | ○ | ○ | | ○ | ○ | |
| | Gadus morhua | ○ | ○ | ○ | ○ | ○ | | |
| | | | ○ | | | | | |
| | Larimichthys crocea | ○ | ○ | ○ | ○ | ○ | ○ | ○ |
| | Lates calcarifer | ○ | ○ | ○ | ○ | ○ | ○ | ○ |
| | Nothobranchius furzeri | | ○ | | | ○ | | |
| | Oreochromis niloticus | ○ | ○ | ○ | ○ | | ○ | |
| | Periophthalmodon schlosseri | | ○ | | | | | |
| | Periophthalmus magnuspinnatus | | ○ | | | | | |
| | Pimephales promelas | | ○ | | | | | |
| | | ○ | ○ | | | ○ | | |
| | Poecilia formosa | ○ | ○ | ○ | ○ | ○ | ○ | |
| | | ○ | ○ | ○ | ○ | ○ | ○ | |
| | Poecilia reticulate | ○ | ○ | ○ | ○ | ○ | ○ | |
| | Sebastes nigrocinctus | | ○ | | | | ○ | |
| | Sebastes rubrivinctus | | ○ | | | | ○ | |
| | Stegastes partitus | ○ | ○ | ○ | ○ | ○ | ○ | ○ |
| | Thunnus orientalis | | ○ | | | ○ | | |
| Sarcopterygii | Latimeria menadoensis | ○ | ○ | | | ○ | | |

*Open circles '○' indicate the presence of the gene. When there is more than one FLREV lineage in a genome, their genomic structures were annotated separately. NviFLERV, AciFLERV, AliFLERV, CoeEFV and DrFV-1 protein sequences were used as probes.
† We identified only one accessory gene (acc) in salamander FLERVs (see also Fig. 2); therefore it is unclear whether it is acc1 or acc2.
‡ Two copies of acc1 genes were found in the FLERV, annotated as acc1a and acc1b (see Fig. 2).

annotated ORF-6 (759 nt, 253 aa), −7 (753 nt, 251 aa) and −8 (702 nt, 234 aa) as an 'acc1a', an 'acc1b' and an 'acc2' gene, respectively. Acc1a and acc1b are found in the same position as bel1, while acc2 in the same position as bel2. The hypothetical proteins of these accessory genes, however, do not exhibit similarity to any known proteins, and/or contain any known conserved domains. Further studies of their functions and structures are required to shed more light on these genes. Similarly, ORF-3 (1,395 nt, 465 aa) and −4 (1,002 nt, 334 aa) proteins, which are much longer than those of ORF-6, −7 and −8, also do not contain any known conserved domains; therefore, it is unclear what they are. Nevertheless, it is known that some retroviruses possess accessory genes between pol and env. For example, the lentiviral human immunodeficiency virus contains accessory genes in this region, vif, vpr and vpu, which are essential for viral replication, assembly and release[22]. We thus hypothesize that ORF-3 and −4 are viral accessory genes, and designated them 'acc3' and 'acc4', respectively. Again, our analyses revealed that other ray-finned fish FLERVs also contain these genes (Table 1, see below), supporting that they are indeed accessory genes of these FV-like viruses.

These gene annotations were then transferred to AliFLERV via protein sequence similarity. By examining the distribution of start and stop codons, we determined AliFLERV to have 6 ORFs (ORF-1 to -6) flanked by 5′- and 3′ LTRs. Our analyses suggest that ORF-1 to -6 are gag, pol, acc3, env, acc1 and acc2 genes, respectively. A $tRNA^{Lys}$-utilising PBS was also identified after the 5′-LTR via sequence similarity as anticipated, similar to those of AciFLERV and most mammalian FVs. Detailed genomic annotations, such as PBS and TSD sequences, gene lengths and locations of the genes on the contigs, are in Supplementary Note 1.

To estimate the age of AciFLERV and AliFLERV, we used the LTR-dating method as described above. Four pairs of AciFLERV paired LTRs were found, and their pairwise distances were estimated to be between 0.2 and 1.1%. Likewise, we obtained two pairs of AliFLERV paired LTRs, and estimated their pairwise distance to be 3.1 and 4.9%. The neutral rate of fish genomic evolution has been reported to be $\sim 1.46 \times 10^{-8}$ substitutions per site per year[23]. Given the estimated pairwise distances and the neutral rate of fish evolution, we inferred the oldest element of AciFLERV and AliFLERV to be 0.377 and 1.68 Myr old,

respectively, which are surprisingly young given the presence of TEs within their protein coding regions and their high copy numbers (at least 9 for AciFLERV and 23 for AliFLERV). This finding indicates that these FLERVs may still be active, and/or that they have been rapidly proliferating by helper infectious retroviruses via complementation *in trans*. Alternatively, it also could be that these groups of young FLERVs represent concomitant germ-line infections of closely related viruses. These young ages, however, should be interpreted with caution, as LTR-dating could severely underestimate the ages of ERVs if LTR gene conversion happens, reducing the divergence between paired LTRs[24]. Nevertheless, this is unlikely as LTR gene conversion had to happen for all six elements to explain the observation. Further analyses revealed that the TEs within these FLERVs (none of which share the same genomic location) belong to TE groups that contain numerous highly similar elements, a strong indication of recent TE bursts. This observation might explain the presence of TEs in AciFLERV and AliFLERV despite their young age.

**Characterization of fragmented FLERVs**. We used the annotated genomes of NviFLERV, AciFLERV and AliFLERV, as well as those of CoeEFV and DrFV-1, to further annotate the genomic structure of other fragmented FLERVs in other vertebrate genomes via protein sequence similarity (Table 1). We were able to detect the presence of NviFLERV-like *gag*, *pol*, *env* and *acc* genes in the genomes of the Japanese fire belly newt (*Cynops pyrrhogaster*) and Iberian ribbed newt (*Pleurodeles waltl*). Phylogenetic analyses showed that Gag, Env and Acc proteins of *C. pyrrhogaster* FLERVs (CpyFLERVs) form two separate clades (Supplementary Fig. 3), confirming the existence of two lineages of CpyFLERVs as initially suggested by the RT phylogenetic analyses. The presence of NviFLERV-like *gag*, *pol* and *env* genes were also detected in the genome of the Ezo salamander (*Hynobius retardatus*), but not the NviFLERV-like *acc* gene. This could be due to the lack of sequence data, genomic deletion, and/or high degree of sequence divergence. Our analyses also detected a NviFLERV-like *gag* gene in the genome of the smooth newt (*Lissotriton vulgaris*), a salamander that was not initially listed as a species containing FV-like RT sequences (Supplementary Table 1). Again, the absence of sequence data and/or genomic deletion might explain this result. Lastly, we could not detect any genes other than *pol* in the western clawed frog (*Xenopus tropicalis*).

Regarding ray-finned fish FLERVs, while our analyses detected AciFLERV/AliFLERV-like *pol* genes in all of the FLERVs (*pol*: 27/27), AciFLERV/AliFLERV-like *gag*, *env* and *acc1* genes could only be detected in about half of the FLERVs (*gag*: 12/27; *env*: 12/27, *acc1*: 12/27), and *acc2* genes in about a fifth of the FLERVs (5/27). (Note that when there are multiple FLERV lineages present in a single genome, the genomic structure of each lineage was annotated separately by keeping track of which contigs the genes were found in.) Interestingly, in addition to the four fish species, our analyses revealed that sablefish (*Anoplopoma fimbria*) also contain two lineages of FLERVs; one with detectable *gag*, *pol*, *acc3*, *env* and *acc2* genes, and another with only a detectable *pol* gene. Sablefish was not listed as a species containing two FLERV lineages in the initial RT analyses as the RT coding region for the former lineage was absent from the database. Another interesting observation was that, when the *acc1* gene was detected, only a single copy was found in all cases, except in AciFLERV. This suggests that the *acc1* duplication in the AciFLERV progenitor occurred very recently, and is lineage-specific. Our analyses confirmed the presence of *acc3* and *acc4* in 14 and 10 ray-finned fish FLERVs,

respectively. The phylogenetic distribution of *acc3* and *acc4* was further examined to shed more light on their evolutionary history (see below). Finally, our analyses detected DrFV-like *gag*, *pol* and *env* genes in the genome of the fathead minnow (*P. promelas*), and CoeEFV-like *gag*, *pol* and *env* genes in the Indonesian coelacanth (*Latimeria menadoensis*), but could not detect any genes other than *pol* in Australian ghostshark (*Callorhinchus milii*). Again, we note that the absence of evidence is not the evidence of absence; the fact that we could not detect some genes in several fish FLERVs could be due to the lack of sequence data, genomic deletion, genetic divergence or indeed the genuine absence of the genes. To distinguish these possibilities requires genomic examination of their exogenous viral counterparts.

**Phylogenetic analyses**. To investigate the phylogenetic relationship between FLERVs and FVs in more detail, we estimated their Bayesian phylogeny based on a Pol protein alignment (Fig. 3). *A. fimbria* FLERV (AfiFLERV) that contains detectable *gag*, *pol*, *acc3*, *env* and *acc2* genes was not included in this analysis because its Pol sequence was extremely short (42 aa after alignment curation). Also, note that the phylogenetic placement of *L. vulgaris* FLERV (LvuFLERV) was determined separately by using a phylogenetic analysis of Gag proteins, as its Pol protein is not available.

Overall, the results from phylogenetic analyses of RT and Pol protein sequences are largely consistent. We found that, as shown by the RT phylogenetic analysis, ray-finned fish, lobe-finned fish and salamander FLERVs, as well as mammalian FVs all form well-supported monophyletic clades (Bayesian posterior probability > 0.99 for all clades), with tetrapod and lobe-finned fish FVs/FLERVs grouping together to the exclusion of ray-finned fish and shark FLERVs (Bayesian posterior probabilities = 0.98). Unlike the RT phylogeny however, the Pol phylogeny shows that shark FLERVs are paraphyletic instead of forming two separate clades; nevertheless, this phylogenetic pattern is not well-supported (Bayesian posterior probability = 0.72), similar to that in the RT phylogeny. Thus, it is still unclear how the progenitor of shark FLERVs interacted with their hosts.

As previously reported[5,6,25], we found strong evidence supporting the co-speciation history of mammals and their FVs (maximum number of co-speciation events inferred = 13/15 (86.67%); random tip mapping: $N = 500$, $P < 0.002$). Unlike in the case of mammals, several cross-species transmissions were found among ray-finned fish (Fig. 3). Despite this observation however, our analyses still showed that ray-finned fish FLERVs co-diverge broadly with their hosts with strong support (maximum number of co-speciation events inferred = 18/27 (66.67%); random tip mapping: $N = 500$, $P < 0.002$). In the case of salamanders, no significant evidence was found for a history of co-speciation with their FV-like viruses (maximum number of co-speciation events inferred = 4/5 (80%); random tip mapping: $N = 500$, $P = 0.094$).

We also mapped the presence of *acc3* and *acc4* onto the phylogeny to determine the history of their acquisition. We found that they are exclusively limited to just one clade of ray-finned fish FLERVs (Fig. 3), suggesting that they were acquired only once and perhaps both at the same time. This result further supports the phylogenetic distinctiveness of the two lineages of ray-finned fish FLERVs. Nonetheless, we noted that *acc3* and *acc4* genes could not be detected in some of the FLERVs in this clade. This could be due to the lack of sequence data or indeed multiple independent gene deletions as indicated by the non-monophyly of the absence of *acc3* and *acc4* genes (Fig. 3).

Regarding the deeper evolutionary history, our analyses show that mammalian FVs are most closely related to

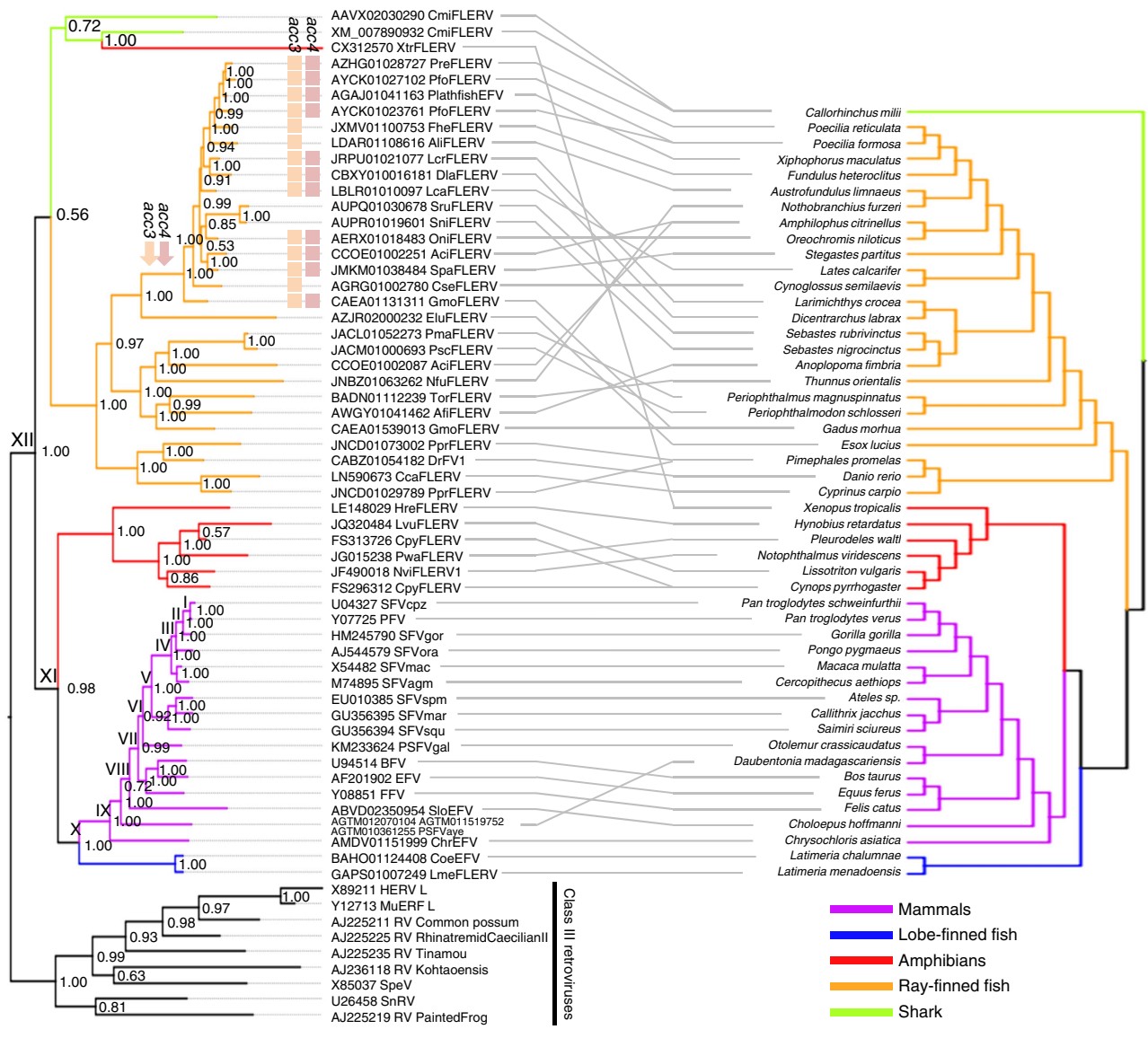

**Figure 3 | Coevolution of FVs and FLERVs and their vertebrate hosts.** A Bayesian phylogeny of FVs and FLERVs (left) is compared with the published vertebrate host cladogram[35-38] (right). Preceding viral names are the contig accession numbers containing viral sequences. Class III retroviruses were used to root the viral tree. Solid grey lines between the two trees indicate viral–host associations. The scale bar (black solid line; underneath the virus phylogeny) represents genetic divergence of length 0.3 in units of amino acid substitutions per site, and Arabic numerals on nodes are Bayesian posterior probability node support values. Roman numerals indicate the nodes of which the total per-lineage substitutions to the chimpanzee simian FV (U14327 SFVcpz) were used to construct the time-dependent rate phenomenon model (Fig. 4). The presence of *acc3* (orange squares) and *acc4* (red squares) is mapped onto the viral phylogeny. The most parsimonious timing of *acc3* and *acc4* acquisition is indicated (orange and pink arrows, respectively).

lobe-finned fish FLERVs (posterior probability > 0.99) and basal to this clade are salamander FLERVs (Bayesian posterior probability > 0.99). This branching pattern, however, does not match that of the hosts, where mammals are more closely related to amphibians than lobe-finned fish. We note that this topology was not supported by the phylogeny estimated from a much shorter RT alignment (130 aa), which shows a sister-taxon relationship between mammalian FVs and salamander FLERVs (Fig. 1), but with no statistical support. Indeed, when the tree estimation uncertainty is taken into account, this phylogenetic pattern falls within the RT tree estimation uncertainty. Combined, our results suggest that there were likely one or more ancient transmissions of FV-like viruses between tetrapods. We also found that, as shown by the RT phylogenetic analysis, instead of clustering with salamander

FLERVs, the frog FLERV (XtrFLERV) was inferred to cluster with shark FLERVs (CmiFLERVs) with strong support (Bayesian posterior probability > 0.99), again indicative of viral cross-class transmission. Nevertheless, interpretation of these findings could be complicated by incomplete lineage sorting. Incomplete lineage sorting is a phenomenon where multiple lineages of viruses continue to exist after the host basal diversification, but are sorted into different host lineages in the subsequent host divergence, and this could give raise to mismatches in virus/host evolutionary histories despite virus-host co-speciation. Temporal evidence can be used to differentiate between this phenomenon and viral cross-class transmissions (see 'Discussion' section). Another possibility is that these mismatches in virus/host evolutionary histories are a result of neutral genetic changes accumulating

within FLERVs. However, a study has shown that neutral genetic changes only increase the branch length and decrease the clade support without altering the tree topology[6]; thus it is unlikely that this is the case. It may nonetheless explain the extremely long branch of XtrFLERV (Fig. 3, left).

**Evolutionary timescale of FLERVs.** Studies have shown that the rate estimates of mammalian FV evolution are time dependent[16,18], and that this time-dependent rate phenomenon (TDRP) can be empirically described well by a power-law decay function[16]. In fact, it has been demonstrated that virtually all viruses exhibit this evolutionary feature, and that the TDRP pattern is extremely stable across a very large timescale, spanning nine orders of magnitude, and across a wide range of host organisms from plants to animals[15]. A consequence of this phenomenon is that the relationship between total per-lineage substitutions ($S$) and evolutionary timescales ($T$) will also be a power-law[16]. To estimate the timescale of the FLERVs, we first built a TDRP model describing the relationship between $S$ and $T$ estimates of mammalian FVs by tracing the chimpanzee SFV lineage backward in time (U14327 SFVcpz; Fig. 3). This was achievable due to the remarkably stable FV-mammal co-speciation history, allowing us to directly infer viral evolutionary timescales from those of mammals, and thereby obtaining a set of corresponding $S$ and $T$ estimates for model construction[16] (Table 2). We then extrapolated the model to estimate the timescale of FLERVs from their $S$ estimates, under an explicit assumption that the TDRP dynamics of mammalian FVs are the same as those infecting their ancient ancestral (perhaps marine) vertebrates (Fig. 4 and Table 2).

Our analyses showed that the model can describe the dynamics of FV substitutions very well (adjusted $R^2 = 0.954$, 95 percent highest probability density interval (95% HPD) = 0.912–0.987). By assuming that viruses infecting modern-day mammals and their ancient vertebrate ancestors share the same TDRP dynamics, and based on the posterior distribution of the Bayesian Pol phylogeny, the clade of mammalian FVs and lobe-finned fish FLERVs was estimated to be ∼263 (95% HPD = 195–342) Myr old. The separation of the salamander FLERV lineage was inferred to have happened ∼348 (95% HPD = 251–478) Ma. The age of the entire clade of FVs/FLERVs was estimated to be ∼455 (95% HPD = 304–684) Myr old.

Since our tree contains both extant retroviruses (which evolve under pure viral evolutionary rates) and ERVs (which have mixed rates of host and virus evolution) (Fig. 3), we note that some

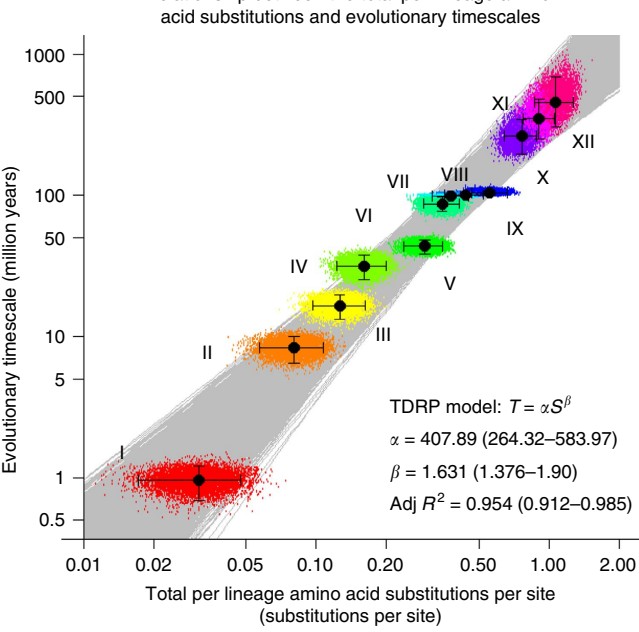

Relationship between the total per-lineage amino acid substitutions and evolutionary timescales

TDRP model: $T = \alpha S^{\beta}$
$\alpha = 407.89\ (264.32{-}583.97)$
$\beta = 1.631\ (1.376{-}1.90)$
Adj $R^2 = 0.954\ (0.912{-}0.985)$

**Figure 4 | Evolutionary timescale of FLERVs estimated by using the power-law rate-decay model.** The total per-lineage amino acid substitutions ($S$) from various nodes to the chimpanzee simian FV (U14327 SFVcpz) are plotted against corresponding evolutionary timescales ($T$). The $S$ and $T$ estimates are labelled with Roman numerals (I–XI), referring to nodes in Fig. 3. Solid black dots are median estimates, and the associated 95% HPDs intervals are indicated by error bars. The $T$ estimates of node I–IX were directly inferred from those of their hosts[35,39,40]. 7,500 power-law TDRP models were fitted to the posterior distributions of the $S$ and $T$ estimates of the nodes I–IX (grey lines). The median model parameter values, adjusted $R^2$ scores, and corresponding 95% HPDs (in the parentheses) are shown in the bottom right. The model was extrapolated to infer the branching date of lobe-finned fish FLERV lineage (node X), the separation date of salamander FLERV lineage (node XI) and the age of the entire FV/FLERV clade (node XII). See Table 2 for the values.

**Table 2 | Evolutionary timescales of FVs and FLERVs.**

| Node* | Total per-lineage substitution (substitutions per site) | | Evolutionary timescale (million years)† | | Ref |
| --- | --- | --- | --- | --- | --- |
| | **Median** | **95% HPD‡** | **Median** | **95% HPD** | |
| I | 0.031 | 0.017–0.048 | 0.96 | 0.69–1.21 | (ref. 39) |
| II | 0.080 | 0.057–0.107 | 8.29 | 6.52–10.05 | (ref. 40) |
| III | 0.126 | 0.097–0.163 | 16.52 | 13.33–19.61 | |
| IV | 0.160 | 0.123–0.199 | 31.60 | 25.36–37.85 | |
| V | 0.292 | 0.238–0.349 | 43.43 | 38.39–48.08 | |
| VI | 0.348 | 0.288–0.413 | 86.92 | 76.34–97.22 | |
| VII | 0.381 | 0.315–0.454 | 98.59 | 95.94–100.78 | (ref. 35) |
| VIII | 0.436 | 0.357–0.523 | 100.95 | 98.40–103.83 | |
| IX | 0.559 | 0.464–0.660 | 103.92 | 99.73–109.09 | |
| X | 0.764 | 0.641–0.889 | 262.76 | 195.00–342.08 | NA |
| XI | 0.908 | 0.771–1.056 | 347.89 | 250.62–479.97 | |
| XII | 1.071 | 0.869–1.274 | 454.65 | 303.69–683.66 | |

*Referring to the node numbers in the viral tree in Fig. 3 (left). Total per-lineage substitution and evolutionary timescale estimates of nodes I–IX were used to estimate the power-law time-dependent rate phenomenon model (Fig. 4). The model was then extrapolated to infer the timescale of nodes X–XII from their total per-lineage substitution estimates.
†Evolutionary timescale of nodes I–IX were directly inferred from those of their hosts. The references refer to the host date references.
‡'95% HPD' = 95% highest probability density interval.

branches in our tree have mixed rates of virus and host evolution, and because our model was built based on evolutionary dynamics of extant viruses, these mixed rates of ERV evolution could potentially bias our date estimates. Nevertheless, since we did not impose a molecular clock onto the tree (see 'Methods' section), each branch can therefore vary their length and height independently of time, allowing them to have different rates. Subsequently, those mixed rates of evolution will tend to limit to terminal branches that lead to ERVs. We avoided these branches with mixed rates of ERV evolution in our date estimation by measuring the node heights from SFVcpz tip down the tree, assuming that internal branches reflect only exogenous viral evolution and do not contain any evolutionary periods in which viruses might have been infectious ERVs.

As an independent verification that our date estimates are not biased by the mixed rates of ERV evolution, we performed phylogenetic analyses and date estimation by using FLERV Pol ancestral sequences, from which the host evolution has been removed. Indeed, we obtained similar results, estimating the divergence date of the lobe-finned fish FLERV lineage to be ∼267 (95% HPD = 150–452) Ma, the separation date of the salamander FLERV lineage to be ∼342 (95% HPD = 157–603) Ma and the overall age of FVs/FLERVs to be ∼473 (95% HPD = 225–897) Myr old (Supplementary Fig. 4). Overall, these similarities in the date estimates suggest that we have successfully avoided the problem of mixed rates of ERV evolution in terminal branches. We note that the date estimate uncertainties are significantly larger than those of our initial date estimates. This is to be expected however, as there were fewer data points available for the TDRP model calibration after the exclusion of endogenous FVs for which ancestral sequences could not be reconstructed due to the lack of data (nine data points in the initial analyses, Fig. 4, compared to six data points in this analysis, Supplementary Fig. 4). In addition to this, it is likely that the ancestral sequence reconstruction may have also introduced some extra uncertainty into the genetic divergence estimation, contributing further to the increase in the date estimate uncertainties.

## Discussion

Here we report 36 lineages of novel FLERVs residing in the genomes of salamanders, a frog, ray-finned fish, lobe-finned fish and shark (Table 1, see also Supplementary Table 1). These FLERVs are phylogenetically basal to mammalian true FVs (Fig. 3), and their functional exogenous viral counterparts are not currently available for direct examination. It is therefore still unclear whether or not we should classify them as FVs. Nonetheless, given the similarity observed between the protein sequences of mammalian FVs and those of salamander/lobe-finned fish FLERVs as well as their phylogenetic positions, we consider that classifying salamander and lobe-finned fish FLERVs as FVs may be appropriate. In contrast, ray-finned fish FLERVs are drastically different from mammalian FVs, both in terms of genome length and gene number. Indeed, to our knowledge, they are the largest known retroviruses ever documented, approximately twice as large as a typical simple retrovirus (∼8–9 kb). Furthermore, none of their proteins, apart from Pol, exhibit strong similarity to those of mammalian FVs. These observations support that ray-finned fish FLERVs should be classified as a separate genus in the subfamily *Spumaretrovirinae*. Although the International Committee on Taxonomy of Viruses requires that a well characterized viral member must be available to establish a new genus, which may require exogenous viruses, we tentatively propose the name for these ray-finned fish FV-like viruses 'Gigantospumavirus'. Our results should stimulate efforts to search for novel retroviruses that might be circulating among modern-day fish and amphibians.

Our results extend the stable and long-term co-speciation history of FV-like viruses and their hosts beyond mammals[5,6,25], showing that the viral progenitors of ray-finned fish FLERVs also co-diverged broadly with their hosts. The co-evolutionary history of ray-finned fish FLERVs, however, is more dynamic than that of mammalian FVs, as indicated by their acquisition of additional accessory genes and that some ray-finned fish genomes even contain two distinct lineages of FLERVs. In the case of salamander FLERVs, we did not find significant evidence supporting an overall virus-host co-speciation; although, since our analyses contained only six lineages of salamander FLERVs, this could be due to the lack of sequence data and/or the low power of the analyses. Overall, our analyses have extended the co-speciation history of this major retroviral lineage from mammalian FVs to cover ray-finned fish FLERVs. To our knowledge, a co-speciation pattern at a scale this large has never been demonstrated in any virus, making this retroviral group a very valuable viral macroevolutionary model.

Our analyses show that FVs and FLERVs from hosts of the same class tend to cluster together, strongly supporting monophyletic clades of mammalian FVs, salamander FLERVs, lobe-finned fish FLERVs and ray-finned fish FLERVs. This indicates that viral transmissions between vertebrates of different classes are rare, consistent with the overall co-speciation history of FVs/FLERVs. Nonetheless, two cases of cross-class viral transmissions were observed. We found that XtrFLERV, a frog FLERV, is placed within the clade of shark FLERVs, CmiFLERVs, with strong support and not with other amphibian (salamander) FLERVs. This finding suggests that the origin of XtrFLERV involved cross-class transmission of a FV-like virus from a shark to an amphibian. Nevertheless, our result does not preclude a possibility that this conflicting virus/host tree topology is a result of long-term incomplete lineage sorting. Thus, this particular finding should be interpreted with caution.

A more robust cross-class viral transmission involves lobe-finned fish FLERVs. Han and Worobey[12] reported the discovery of a lobe-finned fish FLERV, namely CoeEFV. They found that CoeEFV is placed basal to mammalian FVs and that there is a linear correlation (R² = 0.71) between viral and host divergence. Based on these findings, they proposed an ancient co-speciation of CoeEFV and mammalian FVs dating back ∼409 Ma, implying a marine origin of FVs. Nevertheless, since there was only one lobe-finned fish FLERV (ie, CoeEFV) in their study, robust statistical analyses could not be performed to verify the co-speciation, and limited data could obscure a history of viral cross-species transmissions.

At face value, our RT phylogeny seems to suggest that mammalian FVs and salamander FLERVs form a mono-phyletic clade, and lobe-finned fish FLERVs are basal to this clade, consistent with a co-speciation scenario (Fig. 1). However, the support of this phylogenetic pattern is low (bootstrap clade support = 49% and Bayesian posterior probability = 0.88), and thus, it should not be over-interpreted. Indeed, our phylogenetic analyses of longer Pol protein sequences reveal that lobe-finned fish FLERVs are in fact within the clade of mammalian FVs and salamander FLERVs with robust support (posterior probability > 0.99), which in fact is within the uncertainty of the RT phylogeny. On the basis of the well-supported Pol tree topology, we estimated that lobe-finned fish FLERVs diverged from the mammalian FVs ∼263–267 Ma. Overall, this is in conflict with the host diversification pattern, where lobe-finned fish are more basal than amphibians, diverging from mammals ∼409 Ma[26] and ∼335 Ma[27], respectively. Unlike in the case of XtrFLERV, the fact that the divergence

date of lobe-finned fish FLERVs and mammalian FVs is much younger than the hosts' divergence date excludes the possibility of incomplete lineage sorting, and that the viruses truly did not co-speciate with their hosts. Thus, both the phylogenetic placement and temporal origins of CoeEFV are not consistent with co-speciation nor do they provide evidence for a marine origin of FVs. Instead, the branching date of lobe-finned fish FLERVs falls within the period when therapsids—a group of mammal-like reptiles—diversified and dominated the land[28]. Together, our results suggest that lobe-finned fish FV-like viruses do not have a marine origin, but instead originated from one or a series of cross-class transmissions that happened in the middle Permian, ultimately from a prehistoric therapsid to a lobe-finned fish.

On the basis of the Pol phylogeny, we estimated the branching of salamander FLERVs to have happened ∼342–348 Ma, highly comparable to that of their amphibian hosts ∼335 Ma[27], and the age of FVs/FLERVs to be ∼455–473 Myr old, strikingly similar to the timescale of their jawed vertebrate hosts, ∼465 Myr old[27]. Combined with the observed phylogenetic pattern, these date estimates support the hypothesis of a marine origin of FVs. Our results strongly suggest that this group of viruses originated together with their jawed vertebrate hosts in the Ordovician ocean, and underwent a water-to-land evolutionary transition with their hosts, co-evolving with one another for >450 Myr. To our knowledge, this is the oldest date ever directly inferred for any viruses, greater than the age estimate of the oldest known dsDNA viruses by 140 Myr (310 Myr old[29]).

Our results offer key insights into the early history of retroviral evolution as a whole. While retroviruses can be found throughout marine and terrestrial vertebrates, with fish and marine tetrapod ERVs sometimes shown to be basal[1,2], robust temporal and phylogenetic evidence of their ancient origins has been lacking. The discovery of ancient orthologous ERVs has shed some light on the origin of retroviruses. However, the oldest known orthologous ERVs are only 100 Myr old, providing evidence of their existence during the early diversification of mammals[4]. Virus-host co-speciation is another source of information that can be integrated into analyses to infer the age of viruses, but again, the oldest retroviral age estimate that was derived based on virus-host co-speciation pattern is only 100 Myr old[5,6]. Thus, given the ability of retroviruses to jump across distantly related host species[1,2], no robust evidence of the existence of retroviruses prior to 100 Ma has previously been available and they therefore could be considerably younger than their vertebrate hosts.

By integrating multiple sources of data, evaluating the co-speciation history of FVs, FLERVs and their hosts across multiple vertebrate classes and incorporating the TDRP model, we were able to show that this linage of viruses is ∼455–473 Myr old. Since the most recent common ancestor of all retroviruses must be older than that of FVs/FLERVs, our finding provides the first concrete evidence that retroviruses as a whole must be older than jawed vertebrate hosts, extending the oldest age estimate of retroviruses by ∼350 Myr years. Furthermore, we also found that fish FLERVs are positioned towards the root of the tree, pointing towards a marine origin of retroviruses. Together, our analyses provide evidence, both phylogenetically and temporally, that retroviruses emerged together with their vertebrate hosts in the ocean, ∼460–550 Ma[7–9], in the early Palaeozoic Era, if not earlier.

## Methods

**Screening for FV-like sequences.** Five publically accessible nucleotide databases were screened for FV-like endogenous retroviruses (FLERV) by using tBLASTn and the CoeEFV reverse transcriptase (RT) protein as a probe, including

the database of GenBank non-redundant nucleotide sequences (nr), expressed sequence tags (est), high throughput genomic sequences (HTGS), whole-genome shotgun sequences (wgs) and transcriptome shotgun assembly sequences (TSA). The scope of the screening was limited to vertebrates, excluding mammals (taxid: 40674).

The screening was performed in a stepwise manner. In the first iteration, the number of target contigs was set to 50. Frameshift mutations were checked, and the resulting multiple BLAST hits were combined as one. When there were more than three RT sequences on one contig, only the best three (ranked under the default tBLASTn settings) were kept for further analyses. Together with RT sequences of known alpha-, beta-, gamma-, delta- epsilon-, lenti- and spuma-retroviruses, the phylogenetic relationships of these RT sequences were estimated under the maximum-likelihood (ML) framework by using RAxML[30]. The RTREV + CAT amino acid substitution model with 25 per site rate categories was employed to construct the phylogeny. Bootstrap clade support values were calculated using 1,000 pseudoreplicates. We used this RT tree to broadly classify viral sequences into various established retrovirus genera. On the basis of the best estimated ML phylogeny, FV-like RT sequences were identified. The criterion was that they form a monophyletic clade with the RT sequences of mammalian FVs and fish ERVs that have been recognized as FV-like, including CoeEFV, DrFV-1 and platyfishEFV.

To retrieve more RT sequences, the screening process was then repeated with increasing target contig numbers (100, 250, 500 and 1,000) until no additional species were identified as containing FV-like RT sequences. In total, we identified 193 RT sequences as FV-like (Supplementary Table 1). For the final tree, a Bayesian maximum clade credibility phylogeny was also reconstructed to confirm the result, by using MrBayes 3.2.1 (ref. 31). The RTREV + I + Γ(4) amino acid substitution model was used, and no molecular clock was imposed. Two chains of MCMC were run for 1,000,000 steps, and parameters were logged every 1,000 steps with the initial 25% discarded as burn in. A metropolis coupling algorithm (3 hot and 1 cold chains) was applied to improve the sampling. Potential scale reduction factors of all parameters were ∼1.000, indicating that they were all well sampled from their posterior distributions and had converged. The alignment (130 aa, 251 sequences) is available from the authors upon request. The RT tree is shown in Fig. 1 (clade support values are shown only for major clades, and nodes with Bayesian posterior support of <0.5 are not annotated).

On the basis of the identification of LTRs and target site duplications, several potentially full-length FLERVs were found in the eastern newt (*Notophthalmus viridescens*; NviFLERV) and midas cichlid (*Amphilophus citrinellus*; AciFLERV). To search for more FLERVs, we used the RT sequences of NviFLERV, AciFLERV and SloEFV to screen the five databases again as described above. The numbers of the target contigs were set to 100 for the nr, est, HTGS and TSA database screening, and 1,000 for the wgs database screening. These settings were used as they were sufficient to identify all species containing FV-like RT sequences in the first round of screening, in which the RT sequence of CoeEFV was used as a probe. BLAST hits from additional species were added to the phylogenetic analyses to determine whether or not they are FV-like. By doing so, we discovered several more potentially full-length FLERVs in annual killifish (*Austrofundulus limnaeus*; AliFLERV).

**Characterization of novel FV-like elements.** We first characterized NviFLERVs, AciFLERVs and AliFLERVs, which are full-length elements. Since we were able to obtain 9 and 23 elements of AciFLERVs and AliFLERVs, we inferred their ML ancestral sequences by using MEGA 6 (ref. 32) with the GTR + Γ(4) nucleotide substitution model, and used them for genome annotation. The ancestral sequences of the LTRs were inferred based on both 5′- and 3′-LTRs. The alignments used for the ancestral sequence inferences are available from the authors upon request. Potential protein coding regions were identified based on the distribution of stop and start codons, determined by Open Reading Frame Finder (http://www.ncbi.nlm.nih.gov/gorf/orfig.cgi), and were annotated via protein sequence similarity by using BLASTp. Similarity of PBSs to tRNAs was inferred by using the tRNA database (trna.bioinf.uni-leipzig.de). To estimate the age of NviFLERV, AciFLERV and AliFLERV, we used the LTR-dating method. MEGA 6 (ref. 32) was used to calculate the number of nucleotide substitutions between available paired LTRs under the Tamura-Nei DNA evolutionary model, which takes the differences between transition and transversion substitution rates as well as the inequality of nucleotide frequencies into account. We then divided the distance between paired LTRs by two to obtain a total per-lineage LTR substitution estimate, which in turn was divided by the rate of the LTR evolution to derive the time it took to accumulate the observed number of substitutions, ie, their age. We assumed that the LTR rate of evolution is equal to the neutral rate of the host genome. Finally, we used the annotated genomes of NviFLERV, AciFLERV and AliFLERV, as well as those of CoeEFV and DrFV-1 to characterize other FLERVs in other vertebrates (Supplementary Table 1) via protein sequence similarity by using tBLASTn.

**Phylogenetic analyses.** To investigate phylogenetic relationships between FVs and FLERVs in more detail, we estimated their phylogeny based on a manually curated Pol protein alignment (580 aa). Gag and Env proteins were not used because, in the case of amphibian FLERVs, they could not be concatenated with the

Pol proteins as they were found on different contigs in all cases, except only for those of NviEELRV-1. In the case of fish FLERVs, they were not used because they could not be aligned to those of mammalian FVs due to high sequence divergence. As a result, only Pol proteins were used to reconstruct the phylogeny. The alignment contains only one FLERV per monophyletic clade of FLERVs of the same host species, determined by the initial RT phylogenetic analyses. Class III retroviruses were used to root the tree. The alignment is available from the authors upon request. The phylogeny was estimated under the Bayesian framework by using MrBayes 3.2.1 (ref. 31) with the best amino acid substitution model, $LG + I + \Gamma(4) + F$, determined under the AICc criterion by PartitionFinder 1.1.1 (ref. 33). No molecular clock was imposed (that is, a non-clock and unconstrained tree). Two chains of MCMC were run for 1,000,000 steps, and parameters were logged every 1,000 steps with the initial 25% discarded as burn in. A metropolis coupling algorithm (3 hot and 1 cold chains) was applied to improve the MCMC sampling. Potential scale reduction factors of all parameters were ~1.000, indicating that they were all well sampled from their posterior distributions and had converged. Furthermore, since the genome of the smooth newt (*Lissotriton vulgaris*) contains only a Gag coding region (see 'Results' section: Characterization of fragmented FLERVs), it was not included in this analyses. To estimate its phylogenetic position, a separate Gag phylogeny of salamander FLERVs was estimated by using the same method. The best amino acid substitution model was determined to be $WAG + \Gamma(4) + F$ under the AICc criterion by PartitionFinder 1.1.1 (ref. 33). Potential scale reduction factors of all parameters were ~1.000. The alignment of Gag (305 aa) is also available from the authors upon request.

**Co-speciation analyses.** Viral–host co-speciation history was evaluated by comparing the topologies of the viral and host trees by using Jane V4.01 (ref. 34). The analyses were divided into three subanalyses: those of mammalian FVs, ray-finned fish FLERVs and salamander FLERVs. Since there were only two lobe-finned fish and one shark species that contain FLERVs, analyses for these two lineages were not performed. Nodes with <0.80 Bayesian posterior probability support were collapsed with their most adjacent basal node to form a polytomy. The host trees used in these analyses are published host trees[35–38]. The number of co-speciation events was calculated by using a genetic algorithm, with the number of generations, and population set to 100. To maximize the number of co-speciation events inferred, the (vertex-based) costs were set as follows: co-speciation = −1, duplication = 0, duplication and host switch = 0, loss = 0 and failure to diverge = 0. To assess the probability of observing the inferred number of co-speciation events by chance, the random tip mapping method implemented in Jane v4.01 (ref. 34) was used (generation = 100, population = 100 and sample size = 500).

**Estimating evolutionary timescale of FLERVs.** For each of the estimated posterior Bayesian Pol phylogenies, a power-law model describing the relationship between mammalian FV total per-lineage amino acid substitutions ($S$) and evolutionary timescales ($T$) was constructed by tracing the chimpanzee simian FV linage (U14327 SFVcpz) backward in time (Fig. 3, nodes I–IX). The $T$ estimates were directly inferred from those of their hosts[35,39,40] under the well-established FV-host co-speciation assumption[5,6] (Table 2, nodes I–IX). The model fitting was performed by using the *lm* function implemented in R 3.2.1 (ref. 41). The values of $S$ and $T$ were log-transformed (base 10) prior to the linear model fitting. The model was then extrapolated to estimate the timescale of other nodes from their $S$ estimates. This includes the separation date of the lobe-finned fish FLERV lineage (Fig. 3 and Table 2, node X), the separation date of the salamander FLERV lineage (Fig. 3 and Table 2, node XI) and the age of the entire clade of FVs/FLERVs (Fig. 3 and Table 2, node XII). This process was applied to all of the 7,500 posterior Bayesian phylogenies to obtain their full posterior distributions (Fig. 4 and Table 2).

**Phylogenetic analyses by using ML FLERV ancestral sequences.** To assess the effects of the mixed rates of virus–host evolution of FLERVs on our date estimation, we also performed phylogenetic analyses by using ML FLERV ancestral sequences. For a particular FLERV lineage, its *pol* nucleotide ancestral sequence was inferred by using MEGA 6 (ref. 32), with the GTR + $\Gamma(4)$ nucleotide substitution model. The immediate outgroup of the clade was used to root the tree. FLERV lineages that consist of one sequence sample were excluded from analyses as ML ancestral reconstruction requires more than one sequence, unless samples from the same host genus (but different species) exist. In such cases, they would be grouped with the other FLERV lineage and their ancestor would be inferred together. In total, 23 ML FLERV ancestral sequences were reconstructed. Their Pol protein (629 aa) phylogeny, together with those of extant mammalian FVs, was estimated by using MrBayes 3.2.1 (ref. 31) as described above, with the best-fit RTREV + I + $\Gamma(4)$ + F substitution model, determined under the AICc criterion by PartitionFinder 1.1.1 (ref. 33). The alignment is available from the authors upon request. The tree was rooted according to the tree in Fig. 3, left. The position of the root was confirmed by using several Class III retroviruses as outgroups. The TDRP analyses were subsequently performed as described above to estimate the branching

date of the lobe-finned fish FLERV lineage, and that of the salamander FLERV lineage, as well as the age of the entire clade of FVs/FLERVs.

**Data availability.** All data used in this study are available from the authors upon request.

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

## Acknowledgements

P.A. is funded by the Royal Thai Government, A.K. is funded by the Royal Society. The funders had no role in study design, data collection and analysis, decision to publish or preparation of the manuscript. We thank members of the research group for constructive comments and discussions.

## Author contributions

P.A. and A.K. conceived the project, performed the analyses, interpreted the results and wrote the paper.

## Additional information

**Competing financial interests:** The authors declare no competing financial interests.

