## [Peer Review File · Nature Communications]

Reviewers' comments:

Reviewer #1 (Remarks to the Author):

This is an interesting paper, which is however rather confirmatory, since Foamy-like ERVs have been already reported in coelacanth and several species of teleost fish, with a marine origin of these retroviruses having already been proposed. This study extends the analysis to amphibians, other teleost species and one shark species (but only with a pol sequence and no structural information). The paper provides new information on the structural evolution of the viruses, and claims cross-class transmission (but this might be an overstatement).

Please provide more information on the different accessory genes/proteins. Size of the proteins, any conserved domains, any homologous sequences in the genome outside of ERVs? Possible to trace back acc's origins?

P. 5 line 73: 63% is a rather low value, is this grouping supported by other methods of analysis?

P. 5 line 76: the presence of LTRs does not always indicated full-length elements, for instance internal deletions can occur and remove parts of the element between the LTRs.

P. 5 line 80: including genome databases? Precise the types of databases analyzed.

P. 5 line 81: what is SloRFV, define at first use.

Did authors identify TSDs flanking ERV insertions? This is particularly useful to test if the insertion is a bona fide one or a recombinant between two insertions when calculating divergence between LTRs.

P.6 line 97: did you really estimate an "ancestral" or rather a "consensus" sequence?

P.6 line 102: any conservation between mammalian and non-mammalian tRNA PBS sequences?

P. 6 line 106: This E value is low.

P. 7 line 145: It is not possible to claim homology between sequences, which assumes common evolutionary origin, only on the basis of sequence position.

P. 8 line 166: or concomitant germ-line infection by related elements.

P. 11 and abstract. The claim of cross-class transmission, only based on a single phylogenetic analysis, is an overstatement. Alternative hypotheses include the presence of different paralogous retrovirus lineages, with incomplete lineage sorting, or differential evolutionary rates (there is certainly no molecular clocks for these sequences, which have evolved partly as exogenous sequences with no proof-reading correction after reverse transcription. Accordingly, check for instance the suspiciously long branch of XtrFLERV.

P. 15 FLERVs from coelacanth, zebrafish and platyfish must be analyzed with the same methods and included.

Reviewer #2 (Remarks to the Author):

Summary

In this study, a set of foamy virus-like endogenous retroviruses (FLERVs) in fish and amphibian genomes are identified, and the time of divergence of these viruses from the

mammalian foamy virus clade is estimated. The authors show that these novel ERVs are more closely related to modern foamy viruses than to class III 'spuma-like' ERVs, and, like foamy viruses, often have accessory genes, though some of the genes present in these viruses have no sequence homology to known foamy virus accessory genes.

Some of the full length proviruses found are remarkably large, over 17 kb, and contain as many as 5 putative accessory genes. Dating by LTR divergence indicates that these FLERVs are quite young, and the authors suggest that they may still be actively replicating.

Phylogenetic analysis shows that ray-finned fish FLERVs, amphibian FLERVs and mammalian foamy viruses are all monophyletic groups, with one exception, a frog FLERV that groups with shark sequences. The authors suggest that this is the result of an ancient cross-class transmission event; they also argue that coelacanth FLERVs were the result of a cross-class transmission from mammals, as they are more closely related to mammalian foamy viruses than amphibian FLERVs are. The authors looked for evidence of co-speciation within the major foamy virus clades, and find signs of co-speciation for mammalian foamy viruses and ray-finned fish FLERVs, but not salamander FLERVs.

Lastly, the authors build a model of foamy virus divergence rates, calibrating the model with the known divergence times of mammalian foamy viruses. They extrapolate from the mammalian foamy viruses to predict that the mammalian foamy virus lineage diverged from the coelacanth, salamander, and ray-finned fish/shark FLERV clades approximately 263, 348, and 455 million years ago, respectively. As this last date is similar to the estimated age of the jawed vertebrate clade, the authors suggest that foamy viruses originated at the same time as jawed vertebrates, and that retroviruses as a whole are at least this old as well.

Critique and major comments

The two most striking results of this paper are the unusually large, complex proviruses identified in fish, and the estimation that foamy viruses evolved at the same time as jawed vertebrates. The former result is quite well supported, and it will be very interesting if these turn out to be actively replicating.

While the age estimates are interesting and plausible, I'm not entirely confident in the methodology. I'm especially concerned about extrapolating from exogenous foamy virus evolution to infer dates for endogenous sequences. The authors say that 'the model was derived from, and extrapolated to, a lineage of pure viral evolution, and thus there was no need for the adjustment of host neutral evolution,' however the model is based off a tree that includes endogenous sequences, that presumably have evolved as neutral elements in the host genome for considerable periods, interspersed with evolution as viruses for unknown periods of time. The dating relies on an extrapolation of well supported rate of evolution among mammals over a 4-fold greater time scale through the very different marine organisms. Since evolutionary "rates" of endogenous viruses must include both evolution while resident in the genome, and evolution during transmission as a virus, whose proportional contributions are completely unknown, and which might vary considerably from marine to terrestrial environments, and for which the relative rates are completely unknown, it is unclear to me how they adjust for this. One method to avoid this problem is to build a tree using consensus sequences that should represent the exogenous ancestral virus for each ERV lineage, as was done by Diehl et al. in their recent paper on ERV-Fc. It would also be interesting to know how the model does at predicting the ages of nodes VIII and IX, since those nodes include endogenous sequences.

Minor comments

It would be helpful to have a figure showing how these new sequences relate to retroviruses as a whole, something like the unrooted trees in the Han et al. coelacanth ERV paper and the Jern et al. 2005 paper on ERV phylogeny.

p.7, lines 144-146, and p.30, line 618: I'm not sure it makes sense to claim that acc1 and acc2 are 'likely' bel1 and bel2 homologues based purely on their position in the genome. Even if they turn out to have similar functions, some evidence of sequence or structural similarity and/or conservation across all foamy viruses is needed to claim homology; for example, HIV rec and MMTV rem have similar functions and similar positions, but are unlikely to be truly homologous. This issue is reinforced by the fact that the salamander FLERVs appear to have an accessory gene at the same position that may be derived from an entirely different virus, which suggests that it is unlikely to be homologous to the acc1 or acc2 genes from fish.

p.10, lines 217-222: The co-speciation analysis could use some clarification; the authors say that they found evidence for co-speciation in mammals and ray-finned fish, but not salamanders, but without more context it's unclear what the 'potential co-speciation' numbers given mean. It seems clear from the tanglegram that the ray-finned fish viruses are much more prone to cross-species transmission than mammalian foamy viruses; can this be quantified using the Jane software package?

Reviewer #3 (Remarks to the Author):

The manuscript from Aiewsakun and Katzourakis describes the use of foamyvirus-related queries to gather and analyze foamyvirus-like endogenous retrovirus sequences and reconstruct the timing and origins of this lineage of retroviruses. Phylogenetic analysis reveals long-term co-evolution of FV and hosts, but also reveals a number of likely interspecies/interclass transmissions. Modeling suggests that the lineage arose some 450 mya+ as exogenous viruses of marine vertebrates. This places the origins close to the origins of the animal lineages leading to modern vertebrate taxa. In addition, the study has uncovered some unusual retroviral forms, including one that is substantially longer than any known retroviral genome. The manuscript is clearly written and easy to follow.

The analyses are straightforward, and there are no technical concerns. The overall conclusion is not surprising, and in some ways the discussion seems to downplay the fact that it is already widely believed that retroviruses must be at least this old, based on extremely wide distribution among vertebrates (including ERV), absence of evidence in basal vertebrates, close relationship to retrotransposons, etc.

The results don't rule out the possibility that retroviruses are considerably older, or even predate the Paleozoic, so could be misleading to say that the authors have provided direct evidence that retroviruses emerged "with" vertebrate hosts >460 myr ago in the early Paleozoic. For example, the underlying assumption behind using Type III ERVs to root the tree is that retroviruses as a family must be older than the ancestor of all the foamyviruses and FV-like viruses being analyzed here. A common ancestor for all Retroviridae would

predate all of the lineages in figure 2, and in the absence of other data, could be considerably older. Strictly speaking, the authors have only provided evidence for an ancient marine origin of FV-like viruses, and this implies that retroviruses are at least this old (which is logical). However, the evidence for retroviruses as a family is not direct, it's inferred. Of course, we don't know whether retroviruses are significantly older, and what sort of taxa they may have infected, or whether those taxa, if they existed, have modern descendants. It could be helpful in preventing misunderstanding for readers, to include a simple tree figure depicting the three lineages within the Retroviridae, with a branch or two (perhaps a collapsed version of figure 1) showing the relationship of the lineages in this study (FV/Type III) and the other major branches.

324-326 The authors might consider a bit of caution here - the phrase "definitive and direct" may be too strong - the results aren't direct, they are inferred from sequence analysis. The practice in such cases should be to consider the data as being "consistent" or "inconsistent" with a particular hypothesis.

Reviewer's report**Title:** Marine origin of retroviruses in the Early Paleozoic Era**Version:** 1**Date:** 01/07/2016**Reviewer number:** 1**Reviewers' comments:**

This is an interesting paper, which is however rather confirmatory, since Foamy-like ERVs have been already reported in coelacanth and several species of teleost fish, with a marine origin of these retroviruses having already been proposed. This study extends the analysis to amphibians, other teleost species and one shark species (but only with a pol sequence and no structural information). The paper provides new information on the structural evolution of the viruses, and claims cross-class transmission (but this might be an overstatement).

Response: We thank the reviewer for their interest in our paper. We can understand why they might feel parts of it were confirmatory due to the way we structured our introduction and discussion. We did not adequately explain the existing literature or how our work provides a substantial advance over existing work. We disagree that our paper is confirmatory, and hope that the rewritten manuscript fully explains what previous work existed, and also where previous work was limited. The changes can be seen in the tracked document, but we also summarise some of the key points below.

First, while it is true that foamy-like endogenous retroviruses (FLERVs) have been already reported in coelacanth (Han and Worobey, 2012, *PLoS pathogens*) and 3 species of teleost fish (Schartl *et al.*, 2013, *Nat. Genet.*; Llorens *et al.*, 2009, *Biol. Direct*), to the best of our knowledge, amphibian and shark FLERVs have not yet been reported. We also described 27 additional novel lineages of teleost fish FLERVs. Furthermore, previous studies that reported the fish FLERVs described them only partially. For example, they failed to identify that they are the largest known retroviruses (~17.5 kb), and contain 5' putative accessory genes, both of which are unique characteristic features that distinguish them from typical mammalian foamy viruses (FVs). We also found that some fish FLERVs are very young, and thus might still be active, providing a guideline for where to look for novel extant exogenous FV-like viruses.

Second, while it is true that a marine origin of FVs has already been suggested, it was proposed for the wrong reason, and in fact our analysis shows that the coelacanth sequence is inconsistent with co-speciation and ancient marine origins. Han and Worobey (2012, *PLoS Pathogens*) proposed this hypothesis based on an assumption that the time to most recent common ancestor of coelacanth FLERVs and mammalian FVs is the same as that of coelacanth and mammals, ~409 million years. We showed that this is not the case, and that coelacanth FLERVs are likely the result of a cross-class transmission from a mammal-like reptile, a therapsid, to a lobe-finned fish. Nevertheless, our analysis using additional data of FLERVs in shark and other fish genomes, as well as the time-dependent rate phenomenon model, which is a new class of relaxed molecular clock, provides evidence that is consistent with ancient marine origins for retroviruses. We make these points now fully in the discussion.

We also now clarify in the introduction and discussion that retroviruses are widespread among vertebrates, and that there are two competing hypotheses that could explain this pattern. One involved that they are as old as vertebrates, while the other that they are just 100 million years old and that they have transmitted widely among vertebrates. It is true that many researchers do believe that retroviruses are as old as vertebrates, but no direct evidence has to date been presented supporting this claim. We

believe that our work will be of high interest to many researchers, at the very least as a reference resource of a robust retrovirus age estimate.

We have extensively modified our introduction and discussion to accommodate all these points.

Please provide more information on the different accessory genes/proteins. Size of the proteins, any conserved domains, any homologous sequences in the genome outside of ERVs? Possible to trace back acc's origins?

Response: We have now reported in the main text the length of accessory genes and proteins as well as indicated that none of the accessory proteins exhibit similarity to any known proteins, and/or contain any known conserved domains, except those of NviFLERV-1.

Line 127-133 (mark-up 142-149): *“Between the env and 3’-LTR [of NviFLERV-1] is a stretch of uniquely-mapping 492 nt, 164 aa, that does not exhibit similarity to any genes apart from a gene of Porcine reproductive and respiratory syndrome virus (BLASTx: AEQ61854; E = 6×10⁴)... we hypothesise that this nucleotide region might be an accessory gene of the progenitor of NviFLERV-1. Indeed, further analyses revealed that other salamander FLERVs also contain this gene (Table 1...), supporting that it has a viral origin and is not a host gene.”*

Line 169-180 (mark-up 190-204): *“We ... annotated ORF-6 (759 nt, 253 aa), -7 (753 nt, 251 aa), and -8 (702 nt, 234 aa) [of AciFLERV] as an ‘acc1a’, an ‘acc1b’, and an ‘acc2’ gene, respectively. Acc1a and acc1b are found in the same position as bel1, while acc2 in the same position as bel2. The hypothetical proteins of these accessory genes, however, do not exhibit similarity to any known proteins, and/or contain any known conserved domains. ... Similarly, ORF-3 (1,395 nt, 465 aa) and -4 (1,002 nt, 334 aa) proteins [of AciFLERV], which are much longer than those of ORF-6, -7, and -8, also do not contain any known conserved domains... Again, our analyses revealed that other ray-finned fish FLERVs also contain these genes (Table 1...), supporting that they are indeed accessory genes of these FV-like viruses.”*

Lastly, genes of AliFLERV were determined based on sequence similarity to those of AciFLERV. Again no conserved domains were found in its accessory genes.

The detailed annotations of all genes, including accessory genes, such as their length, location on the contigs, and conserved domains are provided in **Supplementary Note 1**. The origin of the accessory genes (*acc3* and *acc4*) are shown in **Fig. 3**, and discussed in lines 257-262.

P. 5 line 73: 63% is a rather low value, is this grouping supported by other methods of analysis?

Response: We appreciate the reviewer’s concern. A clade bootstrap support of 61% is indeed relatively low. However, this clade is the smallest clade that includes all known mammalian FVs and fish FLERVs. We noticed that, in fact, the support for many of the deep clades is very low (**Fig. S1**), indicating that there may be gappy sequences that float from one place to another in the bootstrap resampling trees, reducing the clade support across the entire phylogeny. Furthermore, our original analysis was not a formal test of monophyly, and included a particularly large taxon set that included many very distantly related sequences, thus complicating phylogenetic analysis and potentially reducing support.

To test this hypothesis, we sub-sampled the taxa using representative and relatively complete sequences, and reconstructed the phylogeny again. We found that this indeed increased the support of

the clade significantly (95% bootstrap and 1.00 Bayesian posterior probability clade support, **Fig 1**), supporting our suspicion. To minimise the confusion, we have now removed the original tree from our manuscript, and presented only the down-sampled tree (**Fig 1** and **Fig S1**).

P. 5 line 76: the presence of LTRs does not always indicated full-length elements, for instance internal deletions can occur and remove parts of the element between the LTRs.

Response: We thank the reviewer for pointing this out. The reviewer is correct that the presence of paired LTRs does not always indicate ‘complete’ full-length elements. Indeed, we observed that many of the ERVs that have paired LTRs have large genomic deletions, indicating that they are not complete full-length ERVs. We have now used a softer term, ‘*potentially full-length ERVs*’ instead of ‘*complete full-length ERVs*’ to be more accurate.

Line 82-90 (mark-up 96-104): *“To identify potentially full-length FLERVs, we extended the sequences on both sides... to search for long terminal repeats (LTRs) – the key characteristic feature of ERVs that defines the boundary of the element. By doing so, we discovered one potentially full-length novel FLERV in the eastern newt (Notophthalmus viridescens) and four in the midas cichlid (Amphilophus citrinellus) genomes... Furthermore, we also found five potentially full-length FLERVs in the zebra fish (Danio rerio) and three in the West Indian Ocean coelacanth (Latimeria chalumnae) genomes....”*

Line 93-97 (mark-up 107-111): *“To search for additional FLERVs, we screened the five publically accessible nucleotide databases again... [W]e discovered two more potentially full-length FLERVs in annual killifish (Austrofundulus limnaeus),...”*

Line 113-115 (mark-up 128-131): *“We found that many of the full-length NviFLERV, AciFLERV, and AliFLERV are not completely full-length, containing large insertion-deletion mutations, in-frame stop codon and frameshift mutations as well as transposable elements (TEs), making genome annotation difficult and potentially inaccurate.”*

P. 5 line 80: including genome databases? Precise the types of databases analyzed.

Response: We thank the reviewer for bringing attention to this. The types of databases have now been mentioned in the **Results** section.

Line 73-76 (mark-up 85-89): *“By using tBLASTn and the reverse transcriptase (RT) protein of CoeEFV as a screening probe, 1,752 RT sequences were retrieved from publically accessible nucleotide GenBank databases, including the database of GenBank non-redundant nucleotide sequences, expressed sequence tags, high throughput genomic sequences, whole genome shotgun sequences, and transcriptome shotgun assembly sequences.”*

P. 5 line 81: what is SloRFV, define at first use.

Response: In the restructured introduction, the first mention and definition of SloEFV is on line 51 (mark-up 58).

Did authors identify TSDs flanking ERV insertions? This is particularly useful to test if the insertion is a bona fide one or a recombinant between two insertions when calculating divergence between LTRs.

Response: We thank the reviewer for pointing this out. We were able to identify TSDs of all FLERVs that contain complete paired LTRs. This information has now been added to the manuscript. The details of the TSDs are provided in **Supplementary Note 1**.

Main text

Line 86-89 (mark-up 100-103): “[U]pon the ERV insertion, the target site DNA is duplicated, resulting in small target site duplications (TSDs) flanking the ERVs¹⁹. We were able to identify NviFLERV’s TSDs as well as those of the four AciFLERVs (**Supplementary Note 1**), supporting that they are bona fide ERVs, and not recombinants of multiple elements.”

Line 95-99 (mark-up 109-113): “By expanding the sequences and based on the identification of their LTRs and TSDs, we discovered two more potentially full-length FLERVs in annual killifish (*Austrofundulus limnaeus*), and designated them ‘AliFLERV’. We were able to identify TSDs of only one AliFLERV however, as the contig that harbours the other AliFLERV does not contain complete LTRs (**Supplementary Note 1**).”

Supplementary Note 1

Line 9-10: “Target site duplications (TSDs) [of NviFLERV-1] were also found (5’-GAGT-3’), flanking NviFLERV-1.”

Line 59-61: “We found that the LTRs of AciFLERV are 1,498 nt long (5’-LTR: nt 1-1,498; 3’-LTR: nt 15,912-17,409), and for those with paired LTRs, we were able to identify their TSDs to be 5’-GAAG-3’ (CCOE01001074), 5’-GTGT-3’ (CCOE01000352), 5’-CAGC-3’ (CCOE01001468), and 5’-CTGG-3’ (CCOE01000548).”

Line 102-105: “For one of the two [AliFLERV] elements that has paired LTRs, we found its TSDs to be 5’-ATA[C/A]-3’ (LDAR01108616). The TSDs of the other element could not be identified as the contig that harbours it (LDAR01045383) does not contain complete LTRs (i.e. the 5’ end of the 5’ LTR and the 3’ end of the 3’ LTR are missing).”

P.6 line 97: did you really estimate an "ancestral" or rather a "consensus" sequence?

Response: We apologise for the confusion. We have now modified the text so that it is clearer that we estimated maximum-likelihood ancestral sequences.

Line 117-118 (mark-up 132-134): “[We] reconstructed the maximum-likelihood sequence of the basal node on the phylogeny. This inferred ancestral sequence was used for genome annotation.”

P.6 line 102: any conservation between mammalian and non-mammalian tRNA PBS sequences?

Response: We thank the reviewer for bringing attention to this. NviFLERVs possess a tRNA^{Asn}-utilising PBS similar to the PBS utilised by prosimian galago FV, but differs from those of other mammalian FVs, which are tRNA^{Lys}-utilising PBSs. We also identify fish AciFLERVs, and AliFLERV to have a tRNA^{Lys}-utilising PBS, similar to those of most mammalian FVs. We have now added this information to the main text.

Line 123-125 (mark-up 139-141): “Situated on the 3’ end of the 5’-LTR [of NviFLERV-1] is a tRNA^{Asn}-utilising putative primer binding site (PBS), identified via sequence homology, similar to that of prosimian galago FV⁶, but different from those of other mammalian FVs, which are tRNA^{Lys}-utilising PBSs.”

Line 147-148 (mark-up 164-165): “A tRNA^{Asn}-utilising PBS [of NviFLERV-2] similar to that of NviFLERV-1, and prosimian galago FV, was also identified by via sequence homology.”

Line 165-166 (mark-up 186-187): “Based on sequence similarity and manual sequence inspection, we identified a tRNA^{Lys}-utilising PBS on the 3' end of the 5'-LTR [of AciFLERV], similar to those utilised by most mammalian FVs..”

Line 185-186 (mark-up 210-211): “A tRNA^{Lys}-utilising PBS was also identified after the 5'-LTR [of AliFLERV] via sequence homology as anticipated, similar to those of AciFLERV and most mammalian FVs”

P. 6 line 106: This E value is low.

Response: We appreciate the reviewer’s concern. E value of 6×10^{-4} is indeed too low to indicate a bona fide homology. Nevertheless, together with the fact that the reciprocal blast only returned one hit and the hit is a viral gene, this E value is a good indicator that it is a viral gene rather than a host gene. We have now clarified this in the text.

Line 126-133 (mark-up 142-149): “Between the env and 3'-LTR is a stretch of uniquely-mapping 492 nt, 164 aa, that does not exhibit similarity to any genes apart from a gene of Porcine reproductive and respiratory syndrome virus (BLASTx: AEQ61854; $E = 6 \times 10^{-4}$). This low E value of 6×10^{-4} makes it unlikely that they are homologues; nevertheless since mammalian FVs contain accessory genes in this region (bel1 and bel2), and because it exhibits some similarity to a viral gene, we hypothesise that this nucleotide region might be an accessory gene of the progenitor of NviFLERV-1. Indeed, further analyses revealed that other salamander FLERVs also contain this gene (**Table 1...**), supporting that it has a viral origin and is not a host gene. ”

P. 7 line 145: It is not possible to claim homology between sequences, which assumes common evolutionary origin, only on the basis of sequence position.

Response: We thank the reviewer for pointing this out. Indeed, reviewer #2 also raised the same concern. We have now deleted this claim from the text.

Line 169-173 (mark-up 190-196): “We ... annotated ORF-6 (759 nt, 253 aa), -7 (753 nt, 251 aa), and -8 (702 nt, 234 aa) [of AciFLERV] as an ‘acc1a’, an ‘acc1b’, and an ‘acc2’ gene, respectively. Acc1a and acc1b are found in the same position as bel1, while acc2 in the same position as bel2. The hypothetical proteins of these accessory genes, however, do not exhibit similarity to any known proteins, and/or contain any known conserved domains. Further studies of their functions and structures are required to shed more light on these genes.”

P. 8 line 166: or concomitant germ-line infection by related elements.

Response: We thank the reviewer for this suggestion. We have now added this in our manuscript.

Line 195-198 (mark-up 220-223): “This finding [of young AciFLERVs and AliFLERVs] indicates that these FLERVs may still be active, and/or that they have been rapidly proliferating by helper infectious retroviruses via complementation in trans. Alternatively, it also could be that these groups of young FLERVs represent concomitant germ-line infections of closely related viruses.”

P. 11 and abstract. The claim of cross-class transmission, only based on a single phylogenetic analysis, is an overstatement. Alternative hypotheses include the presence of different paralogous retrovirus lineages, with incomplete lineage sorting, or differential evolutionary rates (there is certainly no molecular clocks for these sequences, which have evolved partly as exogenous sequences with no proof-reading correction after reverse transcription. Accordingly, check for instance the suspiciously long branch of XtrFLERV.

Response: We appreciate the reviewer's concern and thank them for pointing this out. In the case of XtrFLERV, indeed, it is possible that incomplete lineage sorting could be the cause of the mismatch in the virus/host tree topologies, and this has now been discussed in the manuscript [line 347-350 (mark-up 389-392)]. However, in the case of CoeEFV, we believe that incomplete lineage sorting is not the underlying cause of the pattern we observed. Indeed, incomplete lineage sorting can result in mismatching virus/host tree topologies. However, the divergence date of viruses cannot be younger than those of their hosts if they in fact were to co-diverge with one another, otherwise this would mean that CoeEFV has travelled back in time to co-speciate with its host. The fact that we inferred the divergent date of lobe-finned fish FLERVs and mammalian FVs (~263-267 Ma) to be much younger than that of their hosts (~409 Ma) is strong evidence supporting that there was indeed at least one cross-class transmission. Also, our previous study has shown that tree topology is very robust to accumulation of neutral genetic changes; it only increases the branch length and decreases the clade support, but cannot alter the tree topology (Katzourakis et al., 2014, *Retrovirology*). Furthermore, we did not impose a clock when estimating the tree. (We apologise that this information was missing from the previous manuscript; this detail has now been added to **Materials and Methods**). Overall, we thus believe that our claim of cross-class transmission of CoeEFV is not an overstatement and justified. This information has now been added to the text.

Line 271-280 (mark-up 303-313): *"All these findings [of virus-host tree topology conflicts] are indicative of viral cross-class transmissions. However, interpretation of these findings could be complicated by 'incomplete lineage sorting', where multiple lineages of viruses continue to exist after the host basal diversification, but are sorted into different host lineages in the subsequent host divergence, and this could give rise to mismatches in virus/host evolutionary histories despite virus-host co-speciation. Temporal evidence can be used to differentiate between these two scenarios (see **Discussion**). Another possibility is that these mismatches in virus/host evolutionary histories are a result of neutral genetic changes accumulating within FLERVs. However, a study has shown that neutral genetic changes only increase the branch length and decrease the clade support without altering the tree topology⁶; thus it is unlikely that this is the case. It may nonetheless explain the extremely long branch of XtrFLERV, and the poor support of the sister relationship of the shark and ray-finned fish FLERVs (posterior probability = 0.56) (**Fig. 3**, left)"*

Line 347-350 (mark-up 389-392): *"This finding [that XtrFLERV, a frog FLERV, is placed within the clade of shark FLERVs] suggests that the origin of XtrFLERV involved cross-class transmission of a FV-like virus from a shark to an amphibian. Nevertheless, our result does not preclude a possibility that this conflicting virus/host tree topology is a result of long-term incomplete lineage sorting. Thus, this particular finding should be interpreted with caution.*

Line 352-365 (mark-up 394-410): *"A more robust cross-class viral transmission involves lobe-finned fish FLERVs... [W]hile we found that lobe-finned fish FLERVs are basal to mammalian FVs, they are in fact within the clade of mammalian FVs and salamander FLERVs, and diverged from the mammalian FVs ~263-267 Ma. This is in conflict with the host diversification pattern, where lobe-finned fish are more*

basal than amphibians, diverging from mammals ~409 Ma²⁵ and ~335 Ma²⁶, respectively. Unlike in the case of XtrFLERV, the fact that the divergence date of lobe-finned fish FLERVs and mammalian FVs is much younger than the hosts' divergence date excludes the possibility of incomplete lineage sorting, and that the viruses truly did not co-speciate with their hosts. Thus, both the phylogenetic placement and temporal origins of CoeEFV are not consistent with co-speciation nor do they provide evidence for a marine origin of FVs."

P. 15 FLERVs from coelacanth, zebrafish and platyfish must be analyzed with the same methods and included.

Response: As requested, we have now added coelacanth, zebrafish and platyfish genomes in our FLERV searching. The results are in **Fig. S1**. We did not annotate them however, as they have already been described elsewhere (CoeEFV: Han and Worobey, 2012, *PLoS pathogens*; PlatyfishEFV: Scharl *et al.*, 2013, *Nat. Genet.*; and DrFV1: Llorens *et al.*, 2009, *Biol. Direct*), and we do not wish to reproduce these efforts. Nevertheless, CoeEFV, PlatyfishEFV, and DrFV-1 were included in all of our phylogenetic analyses, as well as the FLERV evolutionary timescale estimation.

Reviewer number: 2

Reviewers' comments:

Summary

In this study, a set of foamy virus-like endogenous retroviruses (FLERVs) in fish and amphibian genomes are identified, and the time of divergence of these viruses from the mammalian foamy virus clade is estimated. The authors show that these novel ERVs are more closely related to modern foamy viruses than to class III 'spuma-like' ERVs, and, like foamy viruses, often have accessory genes, though some of the genes present in these viruses have no sequence homology to known foamy virus accessory genes.

Some of the full length proviruses found are remarkably large, over 17 kb, and contain as many as 5 putative accessory genes. Dating by LTR divergence indicates that these FLERVs are quite young, and the authors suggest that they may still be actively replicating.

Phylogenetic analysis shows that ray-finned fish FLERVs, amphibian FLERVs and mammalian foamy viruses are all monophyletic groups, with one exception, a frog FLERV that groups with shark sequences. The authors suggest that this is the result of an ancient cross-class transmission event; they also argue that coelacanth FLERVs were the result of a cross-class transmission from mammals, as they are more closely related to mammalian foamy viruses than amphibian FLERVs are. The authors looked for evidence of co-speciation within the major foamy virus clades, and find signs of co-speciation for mammalian foamy viruses and ray-finned fish FLERVs, but not salamander FLERVs.

Lastly, the authors build a model of foamy virus divergence rates, calibrating the model with the known divergence times of mammalian foamy viruses. They extrapolate from the mammalian foamy viruses to predict that the mammalian foamy virus lineage diverged from the coelacanth, salamander, and ray-finned fish/shark FLERV clades approximately 263, 348, and 455 million years ago, respectively. As this last date is similar to the estimated age of the jawed vertebrate clade, the authors suggest that foamy viruses originated at the same time as jawed vertebrates, and that retroviruses as a whole are at least this old as well.

Critique and major comments

The two most striking results of this paper are the unusually large, complex proviruses identified in fish, and the estimation that foamy viruses evolved at the same time as jawed vertebrates. The former result is quite well supported, and it will be very interesting if these turn out to be actively replicating.

While the age estimates are interesting and plausible, I'm not entirely confident in the methodology. I'm especially concerned about extrapolating from exogenous foamy virus evolution to infer dates for endogenous sequences. The authors say that 'the model was derived from, and extrapolated to, a lineage of pure viral evolution, and thus there was no need for the adjustment of host neutral evolution,' however the model is based off a tree that includes endogenous sequences, that presumably have evolved as neutral elements in the host genome for considerable periods, interspersed with evolution as viruses for unknown periods of time. The dating relies on an extrapolation of well supported rate of evolution among mammals over a 4-fold greater time scale through the very different marine organisms. Since evolutionary "rates" of endogenous viruses must include both evolution while resident in the genome, and evolution during transmission as a virus, whose proportional contributions are completely unknown, and which might vary considerably from marine to terrestrial environments, and for which the relative rates are completely unknown, it is unclear to me how they adjust for this. One method to avoid this problem is to build a tree using consensus sequences that should represent the exogenous ancestral virus for each ERV lineage, as was done by Diehl et al. in their recent paper on

ERV-Fc. It would also be interesting to know how the model does at predicting the ages of nodes VIII and IX, since those nodes include endogenous sequences.

Response: We appreciate the reviewer's concern regarding the age estimation. The reviewer is correct that the tree contains both endogenous and modern-day viruses. We note that we did not impose a clock on to the tree, and thus each branch can vary their height and length with a high degree of freedom. We apologise that this information was missing from the previous manuscript; this detail has now been added to **Materials and Methods**.

It is true that some branches in our tree evolve under a mixed rate of virus and host evolution; however, those branches are limited only to terminal branches leading to the ERVs (as none of the elements are orthologous), and not internal branches, especially the deep ones. To construct the time-dependent rate phenomenon (TDRP) model and extrapolate the model to other parts of the tree, we measured the node height from the SFVcpz tip down the tree, effectively avoiding all branches that contain mixed rates of evolution. For example, regarding the nodes VIII and IX, if we were to measure their height from the SloEFV tip and ChrEFV tip, respectively, then that would be problematic as those branches contain mixed rates of evolution. But by measuring the node heights from the SFVcpz tip, we negated the problem entirely. Furthermore, as can be seen in **Fig 4**, there are no immediate signs that the nodes VIII and IX are outliers and thus should be removed from the model construction. To confirm this, as suggested by the reviewer, we have now performed phylogenetic analyses by using FLERV ancestral sequences, and found that the results we obtained from this analysis are indeed highly similar to those obtained from the original analyses. These results suggest that the effects of the mixed rates of ERV evolution are likely not significant, if any. We have now discussed this thoroughly in our manuscript.

Line 302-313 (mark-up 339-350): *"It is important to note that although our tree contains both extant retroviruses (which evolve under pure viral evolutionary rates) and ERVs (which have mixed rates of host and virus evolution) (Fig. 3), this should not affect our date estimation. This is because we did not impose a molecular clock when estimating the tree (see **Materials and Methods**), and thus each branch can vary their length and height with a high degree of freedom. Some branches in our tree contain a mixed rate of virus and host evolution; however, these branches are limited only to terminal branches leading to ERVs, and not internal branches, especially the deep ones. By measuring the node heights from SFVcpz tip down the tree, we effectively avoided all the branches that contain mixed rates of evolution. Indeed, we obtained similar results from analyses that used FLERV ancestral sequences, estimating the divergence date of lobe-finned fish FLERVs to be ~267 (95% HPD = 150-452) [VS ~263 (95% HPD = 195-342)] Ma, the separation date of the salamander FLERV lineage to be ~342 (95% HPD = 157-603) [VS ~348 (95% HPD = 251-478)] Ma, and the overall age of FVs/FLERVs to be ~473 (95% HPD = 225-897) [VS ~455 (95% HPD = 304-684)] myr old (**Supplementary Fig. 4**). Overall, these findings suggest that the effects of the mixed rates of ERV evolution are likely not significant, if any."*

We also understand the reviewer's concern regarding the model extrapolation over a large timescale. Interestingly, we recently showed that the TDRP pattern is extremely stable across different timescales (spanning 9 orders of magnitude), and viral types (demonstrated in 6 viral Baltimore classification groups) (Aiewsakun and Katzourakis, 2016, *JVI*). Given these observations, we believe that a 4-fold timescale extrapolation from FVs to FLERVs is justified. This is now discussed in our manuscript.

Line 284-294 (mark-up 317-329): *"[I]t has been demonstrated that virtually all viruses exhibit this evolutionary feature, and that the TDRP pattern is extremely stable across a very large timescale,*

spanning 9 orders of magnitude¹⁵...Given that the TDRP pattern is extremely stable across different timescales and viral types¹⁵, we believe that the model extrapolation from FVs to FLERVs is justified"

Minor comments

It would be helpful to have a figure showing how these new sequences relate to retroviruses as a whole, something like the unrooted trees in the Han et al. coelacanth ERV paper and the Jern et al. 2005 paper on ERV phylogeny.

Response: As requested, we have now added the figure to the main text (**Fig. 1**).

p.7, lines 144-146, and p.30, line 618: I'm not sure it makes sense to claim that acc1 and acc2 are 'likely' bel1 and bel2 homologues based purely on their position in the genome. Even if they turn out to have similar functions, some evidence of sequence or structural similarity and/or conservation across all foamy viruses is needed to claim homology; for example, HIV rec and MMTV rem have similar functions and similar positions, but are unlikely to be truly homologous. This issue is reinforced by the fact that the salamander FLERVs appear to have an accessory gene at the same position that may be derived from an entirely different virus, which suggests that it is unlikely to be homologous to the acc1 or acc2 genes from fish.

Response: We thank the reviewer for pointing this out. This issue was also raised by reviewer #1. We have now removed this claim from our manuscript.

Line 169-173 (mark-up 190-196): *"We ... annotated ORF-6 (759 nt, 253 aa), -7 (753 nt, 251 aa), and -8 (702 nt, 234 aa) [of AciFLERV] as an 'acc1a', an 'acc1b', and an 'acc2' gene, respectively. Acc1a and acc1b are found in the same position as bel1, while acc2 in the same position as bel2. The hypothetical proteins of these accessory genes, however, do not exhibit similarity to any known proteins, and/or contain any known conserved domains. Further studies of their functions and structures are required to shed more light on these genes."*

p.10, lines 217-222: The co-speciation analysis could use some clarification; the authors say that they found evidence for co-speciation in mammals and ray-finned fish, but not salamanders, but without more context it's unclear what the 'potential co-speciation' numbers given mean. It seems clear from the tanglegram that the ray-finned fish viruses are much more prone to cross-species transmission than mammalian foamy viruses; can this be quantified using the Jane software package?

Response: We apologise for the confusion. The 'potential co-speciation number' is the maximum number of co-speciation events that can be inferred. To avoid confusion, we have now changed the term to be *'maximum number of co-speciation events inferred'* [line 247-255 (mark-up 274-284)]. The information about the Jane setting is in **Materials and Methods**.

Our analyses indeed clearly showed that fish FLERVs are more prone to cross-species transmissions than mammalian FVs; however Jane offers a 'random tip mapping test', evaluating the inferred number of co-speciation events whether or not it is significantly different from those obtained when the virus-host associations are randomised. Our analyses inferred a maximum number of 18 co-speciation events out of 27 divergence events (66.67%) among fish FLERVs and their hosts, and this is significantly different from those obtained under random tip mapping scenarios (sample size = 500, $p < 0.002$). This result is indicative of a broad co-speciation pattern between fish and their FLERVs, although it is by no means as

strong as those of mammals (13 co-speciation events out of 15 divergence events, 86.67%). This has now been made clearer in the text.

Line 248-253 (mark-up 275-282): *“As previously reported^{5,6,24}, we found strong evidence supporting the co-speciation history of mammals and their FVs (maximum number of co-speciation events inferred = 13/15 (86.67%) ; random tip mapping: N = 500, p < 0.002). Unlike in the case of mammals, several cross-species transmissions were found among ray finned fish (Fig. 3). Despite this observation however, our analyses still showed that ray-finned fish FLERVs co-diverge broadly with their hosts with strong support (maximum number of co-speciation events inferred = 18/27 (66.67%); random tip mapping: N = 500, p < 0.002).”*

Line 331-340 (mark-up 368-382): *“Our results extend the stable and long-term co-speciation history of FV-like viruses and their hosts beyond mammals^{5,6,24}, showing that the viral progenitors of ray-finned fish FLERVs also co-diverged broadly with their hosts. The co-evolutionary history of ray-finned fish FLERVs however is more dynamic than that of mammalian FVs, as indicated by their acquisition of additional accessory genes and that some ray-finned fish genomes even contain two distinct lineages of FLERVs... Overall, our analyses have extended the co-speciation history of this major retroviral lineage from mammalian FVs to cover ray-finned fish FLERVs. To our knowledge, a co-speciation pattern at a scale this large has never been demonstrated in any virus, making this retroviral group a very valuable viral macroevolutionary model.”*

Reviewer number: 3

Reviewers' comments:

The manuscript from Aiewsakun and Katzourakis describes the use of foamyvirus-related queries to gather and analyze foamyvirus-like endogenous retrovirus sequences and reconstruct the timing and origins of this lineage of retroviruses. Phylogenetic analysis reveals long-term co-evolution of FV and hosts, but also reveals a number of likely interspecies/interclass transmissions. Modeling suggests that the lineage arose some 450 mya+ as exogenous viruses of marine vertebrates. This places the origins close to the origins of the animal lineages leading to modern vertebrate taxa. In addition, the study has uncovered some unusual retroviral forms, including one that is substantially longer than any known retroviral genome. The manuscript is clearly written and easy to follow.

The analyses are straightforward, and there are no technical concerns. The overall conclusion is not surprising, and in some ways the discussion seems to downplay the fact that it is already widely believed that retroviruses must be at least this old, based on extremely wide distribution among vertebrates (including ERV), absence of evidence in basal vertebrates, close relationship to retrotransposons, etc.

Response: A similar concern has been raised by reviewer #1. Please see the reply above (comment #1). We have extensively modified the introduction and discussion to make clearer what is known and what has been speculated about the origin of retroviruses, what the difficulties are in estimating its date, and what we have to offer. We do understand why the previous manuscript may have not been clear enough about the novelty of this work relative to the existing literature, and hope the revised version clarifies these issues. There are two competing interpretations of the wide distribution of retroviruses among vertebrates, and our work allows one to be formally ruled out in support of their ancient marine origins.

The results don't rule out the possibility that retroviruses are considerably older, or even predate the Paleozoic, so could be misleading to say that the authors have provided direct evidence that retroviruses emerged "with" vertebrate hosts >460 myr ago in the early Paleozoic. For example, the underlying assumption behind using Type III ERVs to root the tree is that retroviruses as a family must be older than the ancestor of all the foamyviruses and FV-like viruses being analyzed here. A common ancestor for all Retroviridae would predate all of the lineages in figure 2, and in the absence of other data, could be considerably older. Strictly speaking, the authors have only provided evidence for an ancient marine origin of FV-like viruses, and this implies that retroviruses are at least this old (which is logical). However, the evidence for retroviruses as a family is not direct, it's inferred. Of course, we don't know whether retroviruses are significantly older, and what sort of taxa they may have infected, or whether those taxa, if they existed, have modern descendants. It could be helpful in preventing misunderstanding for readers, to include a simple tree figure depicting the three lineages within the Retroviridae, with a branch or two (perhaps a collapsed version of figure 1) showing the relationship of the lineages in this study (FV/Type III) and the other major branches.

Response: Yes, this is correct, they might be older. We have now acknowledged this in our manuscript to clarify any ambiguity about this claim.

Line 391-395 (mark-up 443-449): *"By integrating multiple sources of data together, evaluating the co-speciation history of FVs, FLERVs, and their hosts across multiple vertebrate classes and incorporating the TDRP model, we were able to show that this lineage of viruses is ~455-473 myr old. Since the most recent common ancestor of all retroviruses must be older than that of FVs/FLERVs, our finding provides*

the first concrete evidence that retroviruses as a whole must be older than jawed vertebrate hosts, extending the oldest age estimate of retroviruses by ~350 myr years."

With respect to the suggested tree, we have now added it to our manuscript (**Fig. 1**) as suggested.

324-326 The authors might consider a bit of caution here - the phrase "definitive and direct" may be too strong - the results aren't direct, they are inferred from sequence analysis. The practice in such cases should be to consider the data as being "consistent" or "inconsistent" with a particular hypothesis.

Response: We appreciate the reviewer's concern and thank them for pointing this out. We have now modified the text to be more precise as suggested. The previous version of our manuscript did not make clear enough the ways in which our analysis is far more robust and conclusive than any previous suggestive attempts. In line with the reviewers first comment as well, we hope the modified introduction and discussion clarify these issues. We have changed the language regarding the nature of our evidence and no longer say that it is definitive and direct. Rather, we explicitly describe the nature of our evidence, which is both phylogenetic and temporal.

Line 391-398 (mark-up 443-453): *"By integrating multiple sources of data together, evaluating the co-speciation history of FVs, FLERVs, and their hosts across multiple vertebrate classes and incorporating the TDRP model, we were able to show that this lineage of viruses is ~455-473 myr old. Since the most recent common ancestor of all retroviruses must be older than that of FVs/FLERVs, our finding provides the first concrete evidence that retroviruses as a whole must be older than jawed vertebrate hosts, extending the oldest age estimate of retroviruses by ~350 myr years. Furthermore, we also found that fish FLERVs are positioned towards the root of the tree, pointing towards a marine origin of retroviruses. Together, our analyses provide evidence, both phylogenetically and temporally, that retroviruses emerged together with their vertebrate hosts in the ocean, ~460-550 Ma⁷⁻⁹, in the early Paleozoic era, if not earlier."*

Reviewers' comments:

Reviewer #1 (Remarks to the Author):

In my opinion authors have answered the comments of the reviewers a satisfactory manner.

Reviewer #2 (Remarks to the Author):

While the revised article has addressed most of the concerns we raised previously and has introduced new analyses where requested, the new figures have some serious discrepancies with the original data.

Most importantly, one of the major results of the manuscript is that the lobe-finned fish FLERVs are more closely related to the mammalian foamy viruses than the salamander FLERVs are, thus suggesting a cross-class transmission event, as shown by the trees in figures 3 and S4. However, the trees in figures 1 and S1 do not support this claim, but rather show salamander and mammalian viruses forming a monophyletic clade, with the lobe-finned fish viruses as the sister group, as expected in a co-speciation scenario, and as posited in the original coelacanth FLERV paper. The bootstrap support for the mammalian clade and the mammalian-salamander clade is quite low, which could mean that the original trees were in fact correct, but the discrepancy must be addressed. A second, less important difference between the trees is seen with the shark FLERVs, which are monophyletic in figures 3 and S4, but form two separate clades in figures 1 and S1.

Secondly, the discordant tree topologies cast some doubt on the TDRP age estimates that were derived from the original tree. Clearly the estimates that the salamander viruses diverged earlier than the lobe-finned fish viruses would not make sense if the new trees are correct in placing the lobe-finned fish clade basal to the salamander viruses. While it is encouraging that reanalysis using predicted ancestral sequences produced roughly the same age estimates as the original analysis, it is also somewhat concerning that those estimates have much larger credibility intervals than the original analysis (e.g., the salamander FV divergence date range changed from 251-478 million years ago to 157-603 million years ago).

Finally, I still have concerns about the degree of certainty with which the authors express their principal conclusion. While they have strengthened (or at least clarified) their argument for a complex clock like behavior of virus evolution over 10s of billions of replication cycles, I still don't see the justification for assuming that all the conditions which affect the rates of this clock are the same in terrestrial mammals as in marine fishes. Further, the assumption that the internal branches on the trees reflect only replication as virus and do not include lengthy and variable periods of time as infectious ERVs is not, in my opinion, justified. This study is certainly worth publishing, but the authors need to soften their conclusions, add more caveats, and state their assumptions more explicitly, so the readers can see for themselves the basis for the conclusions drawn.

Reviewer #3 (Remarks to the Author):

My concerns with the original manuscript were largely about exercising caution in presenting what remain indirect conclusions, and some suggestions to make the story easier to follow. After careful re-reading of the manuscript and the authors responses, together with some of the responses to reviewer 1, I feel that these concerns have been addressed.

Reviewer's report**Title:** Marine origin of retroviruses in the Early Paleozoic Era**Version:** 2**Date:** 31/08/2016**Reviewer number:** 2**Reviewers' comments:**

While the revised article has addressed most of the concerns we raised previously and has introduced new analyses where requested, the new figures have some serious discrepancies with the original data.

Most importantly, one of the major results of the manuscript is that the lobe-finned fish FLERVs are more closely related to the mammalian foamy viruses than the salamander FLERVs are, thus suggesting a cross-class transmission event, as shown by the trees in figures 3 and S4. However, the trees in figures 1 and S1 do not support this claim, but rather show salamander and mammalian viruses forming a monophyletic clade, with the lobe-finned fish viruses as the sister group, as expected in a co-speciation scenario, and as posited in the original coelacanth FLERV paper. The bootstrap support for the mammalian clade and the mammalian-salamander clade is quite low, which could mean that the original trees were in fact correct, but the discrepancy must be addressed. A second, less important difference between the trees is seen with the shark FLERVs, which are monophyletic in figures 3 and S4, but form two separate clades in figures 1 and S1.

Secondly, the discordant tree topologies cast some doubt on the TDRP age estimates that were derived from the original tree. Clearly the estimates that the salamander viruses diverged earlier than the lobe-finned fish viruses would not make sense if the new trees are correct in placing the lobe-finned fish clade basal to the salamander viruses.

Response: We thank the reviewer for their interest in our paper, and apologise for the confusion. We can understand why it might seem as if there are discrepancies between the original data and the revised manuscript, due to the way the data are presented and figures moved from supplementary material to the main text. In particular, the current **Fig 1** and **S1**—the RT tree—were in fact present in our original manuscript as **Fig S1** (see **Fig S1** in our original manuscript, NCOMMS-16-11274). We moved it to the main text, as per Reviewer #2's and #3's request for a figure showing how FLERVs relate to other retroviruses as a whole. There are some minor topological differences between our current RT tree and the previous tree however, likely due to the inclusion of additional sequences of coelacanth, zebrafish and platyfish FLERVs, and the removal of 'gappy' sequences from the tree, in order to respond to Reviewer #1's comments. Nevertheless, the previous and current RT trees show the same phylogenetic pattern regarding the relationship between mammalian FVs, lobe-finned fish FLERVs and salamander FLERVs—mammalian FVs appear to cluster with salamander FLERVs and not lobe-finned fish FLERVs, but with no statistical support. We therefore would like to point out that there are in fact no discrepancies between the original results and the ones presented in our revised manuscript.

Nevertheless, as the reviewer pointed out, we did not adequately compare the RT tree and the Pol tree in our previous manuscript, nor did we explain why both trees were reconstructed and presented. This was because the two trees serve different purposes—the RT tree was used for high level classification, in order to identify the monophyletic lineage of FLERVs against a backdrop of known retroviral diversity, whereas the Pol tree was used to investigate detailed phylogenetic relationships among FVs and FLERVs. RT has previously been shown to be sufficient for reconstructing broad relationships between higher-level retroviral taxa, but not for accurately reconstructing more fine scale relationships

between and within retroviral genera. Full-length polymerase alignments are commonly used for the latter purpose as they have much more information and are suitably conserved. We apologise that we did not explain this in our previous manuscripts. We have now explicitly stated in our manuscript that *'We used this RT tree to broadly classify viral sequences into various established retrovirus genera. Based on the best estimated ML phylogeny, FV-like RT sequences were identified. The criterion was that they form a monophyletic clade with the RT sequences of mammalian FVs and fish ERVs that have been recognised as FV-like, including CoeEFV, DrFV-1, and playtyfishEFV'* [Line 464-467 (marked-up 476-479): **Materials and Methods**] and that *'[t]o investigate phylogenetic relationships between FVs and FLERVs in more detail, we estimated their phylogeny based on a manually-curated Pol protein alignment...'* [Line 511-512 (marked-up 523-524): **Materials and Methods**].

As requested by the review, we also have now addressed the discrepancies between the RT tree and the Pol tree in our manuscript in detail, both about the relationship between mammalian FVs, lobe-finned fish FLERVs and salamander FLERVs, and the relationship among shark FLERVs. It is important to note that, since the RT tree was reconstructed based on a short protein sequence alignment (130 aa), many nodes in the tree are not well-supported and hence should not be over interpreted. This is a common practise (Katzourakis et al., *PNAS*, 2007; Katzourakis et al., *Science*, 2009) and thus it should not be used beyond a broad classification of retroviruses. In particular, as noted by the reviewer, *'the bootstrap support for... the mammalian-salamander clade is quite low'*, which could mean that they might in fact not form a clade. Our Pol phylogenetic analysis showed that, indeed, they do not; instead, mammalian FVs and lobe-finned fish FLERVs form a clade, and basal to this clade are salamander FLERVs with strong support. At face value, the RT tree and the Pol tree might appear to conflict with one another, but actually they do not—the two trees are completely consistent when the tree estimation uncertainty and differing amounts of evolutionary information are taken into account. We have now discussed this issue in detail in our manuscript.

Line 105-121 (marked-up 106-122) (Results): *'Our RT phylogeny strongly supports monophyletic clades of salamander, lobe-finned fish, and ray-finned fish FLERVs (bootstrap clade support > 84%, and Bayesian posterior probability = 1.00 for all clades). The clade of mammalian FVs was strongly supported by the Bayesian phylogenetic analysis (Bayesian posterior probability = 0.99), but not by the maximum-likelihood method (bootstrap clade support = 57%). On the other hand, we found that the frog FLERV clusters together with shark FLERVs (bootstrap clade support = 74%, and Bayesian posterior probability = 1.00), and the shark FLERVs appear to form two separate clades; however, the latter phylogenetic pattern is not significantly supported (bootstrap clade support = 38%, Bayesian posterior probability = 0.73). We also observed that mammalian FVs, salamander FLERVs, and lobe-finned fish FLERVs cluster together (bootstrap clade support = 90%, and Bayesian posterior probability = 1.00), and ray-finned fish and shark FLERVs are basal to this clade, reflecting the host history. The estimated RT phylogeny also shows that mammalian FVs are more closely related to salamander FLERVs than lobe-finned fish FLERVs, mirroring the host evolutionary relationship and thus consistent with a long-term co-speciation history between this retroviral lineage with their tetrapod hosts. However, again, the support for this relationship is low (bootstrap clade support = 49%, Bayesian posterior probability = 0.88). These low phylogenetic support values are likely because the RT sequences used to reconstruct the tree were short (130 aa), and because of this, the topology of this tree should not be over interpreted. Indeed, as noted by others^{5,20}, it is suitable only for broad classification of viruses. Below, we reconstructed a phylogeny of longer Pol protein sequences in order to determine the relationship among FVs and FLERVs more precisely (see **Results: Phylogenetic analyses**).'*

Line 266-273 (marked-up 267-274) (Results): *‘Overall, the results from phylogenetic analyses of RT and Pol protein sequences are largely consistent. We found that, as shown by the RT phylogenetic analysis, ray-finned fish, lobe-finned fish, and salamander FLERVs, as well as mammalian FVs all form well-supported monophyletic clades (Bayesian posterior probabilities > 0.99 for all clades), with ray-finned fish and shark FLERVs being basal to tetrapod FVs/FLERVs (Bayesian posterior probabilities = 0.98). Unlike the RT phylogeny however, the Pol phylogeny shows that shark FLERVs are monophyletic instead of forming two separate clades; nevertheless, this phylogenetic pattern is not well-supported (Bayesian posterior probabilities = 0.72), similar to that in the RT phylogeny. Thus, it is still unclear how the progenitor of shark FLERVs interacted with their hosts.’*

Line 291-301 (marked-up 292-304) (Results): *‘Regarding the deeper evolutionary history, our [Pol phylogenetic] analyses show that mammalian FVs are most closely related to lobe-finned fish FLERVs (posterior probability > 0.99), and basal to this clade are salamander FLERVs (Bayesian posterior probability > 0.99). This branching pattern, however, does not match that of the hosts, where mammals are more closely related to amphibians than lobe-finned fish. We note that, this topology was not supported by the phylogeny from a much shorter RT alignment (130 aa), which shows a sister-taxon relationship between mammalian FVs and salamander FLERVs (Fig. 1), but with no statistical support. Indeed, when the tree estimation uncertainty is taken into account, this phylogenetic pattern falls within the RT tree estimation uncertainty. Combined, our results suggest that there were likely one or more ancient transmissions of FV-like viruses between tetrapods. We also found that, as shown by the RT phylogenetic analysis, instead of clustering with salamander FLERVs, the frog FLERV (XtrFLERV) was inferred to cluster with shark FLERVs (CmiFLERVs) with strong support (Bayesian posterior probability > 0.99), again indicative of viral cross-class transmissions.’*

Line 403-408 (marked-up 414-420) (Discussion): *‘At face value, our RT phylogeny seems to suggest that mammalian FVs and salamander FLERVs form a monophyletic clade, and lobe-finned fish FLERVs are basal to this clade, consistent with a co-speciation scenario (Fig. 1). However, the support of this phylogenetic pattern is low (bootstrap clade support = 49%, Bayesian posterior probability = 0.88), and thus, it should not be over interpreted. Indeed, our phylogenetic analyses of longer Pol protein sequences reveal that lobe-finned fish FLERVs are in fact within the clade of mammalian FVs and salamander FLERVs with robust support (posterior probability > 0.99), which in fact is within the uncertainty of the RT phylogeny.’*

Finally, the reviewer correctly pointed out that *‘the estimates that the salamander viruses diverged earlier than the lobe-finned fish viruses would not make sense if the new [RT] trees are correct in placing the lobe-finned fish clade basal to the salamander viruses’*. However, as we now explain above and have also now stated in the manuscript, the RT phylogeny should not be interpreted beyond a broad classification of viruses, especially when the topology is not well-supported. It is a common practise that, in order to investigate detailed phylogenetic relationships of retroviruses, phylogenies estimated from longer sequence alignments will be used instead. In our case, it is the Pol phylogeny. Therefore, to minimise the chance of confusion and in order to not distract the readers, we prefer not to discuss that our date estimates would not make sense if the un-supported relationship between mammalian FVs, salamander FLERVs, and lobe-finned fish FLERVs, estimated from a short RT alignment, were to be correct; however this could be done if this is required. Nevertheless, in order to make clearer to the reader, we have now explicitly stated in various places in our manuscript that our date estimates are based on the Pol phylogeny.

Line 328-330 (marked-up 334-336) (Results): *'...based on the posterior distribution of the Bayesian Pol phylogeny, the clade of mammalian FVs and lobe-finned fish FLERVs was estimated to be...'*

Line 408-409 (marked-up 420-421) (Discussion): *'Based on the well-supported Pol tree topology, we estimated that lobe-finned fish FLERVs diverged from the mammalian FVs...'*

Line 421 (marked-up 433) (Discussion): *'Based on the Pol phylogeny, we estimated the branching of salamander FLERVs to happen...'*

Line 545-547 (marked-up 557-559) (Materials and Methods): *'For each of the estimated posterior Bayesian Pol phylogenies, a power-law model describing the relationship between mammalian FV total per-lineage amino acid substitutions (S) and evolutionary timescales (T) was constructed...'*

While it is encouraging that reanalysis using predicted ancestral sequences produced roughly the same age estimates as the original analysis, it is also somewhat concerning that those estimates have much larger credibility intervals than the original analysis (e.g., the salamander FV divergence date range changed from 251-478 million years ago to 157-603 million years ago).

Response: We thank the reviewer for pointing this out. The increase in date uncertainties is likely due to the fact that there were fewer data points available for the TDRP model calibration in the FLERV ancestral sequence analysis. On top of this, FLERV ancestral sequence reconstruction may also introduce some extra uncertainty in the divergence estimates. This information has now been added into our manuscript.

Line 351-356 (marked-up 361-367) (Results): *'We note that the date estimate uncertainties are significantly larger than those of our initial date estimates. This is to be expected however, as there were fewer data points available for the TDRP model calibration after the exclusion of endogenous FVs for which ancestral sequences could not be reconstructed due to the lack of data (9 data points in the initial analyses, **Fig. 4**, compared to 6 data points in this analysis, **Supplementary Fig. 4**). In addition to this, it is likely that the ancestral sequence reconstruction may have introduced some extra uncertainty into the genetic divergence estimation also, contributing further to the increase in the date estimate uncertainties.'*

Finally, I still have concerns about the degree of certainty with which the authors express their principal conclusion. While they have strengthened (or at least clarified) their argument for a complex clock like behavior of virus evolution over 10s of billions of replication cycles, I still don't see the justification for assuming that all the conditions which affect the rates of this clock are the same in terrestrial mammals as in marine fishes. Further, the assumption that the internal branches on the trees reflect only replication as virus and do not include lengthy and variable periods of time as infectious ERVs is not, in my opinion, justified. This study is certainly worth publishing, but the authors need to soften their conclusions, add more caveats, and state their assumptions more explicitly, so the readers can see for themselves the basis for the conclusions drawn.

Response: We would like to thank the reviewer for expressing their interest in our paper. Indeed, the reviewer is correct that our interpretation of the results and analyses are under the assumption that (i) we can extrapolate the evolutionary dynamics of viruses infecting terrestrial mammals to those infecting their ancient ancestral vertebrates, and (ii) that the internal branches on the trees reflect only

replication as virus and do not include lengthy and variable periods of time as infectious ERVs. We have now explicitly stated these two assumptions in our manuscript as suggested.

Line 321-324 (marked-up 326-329) (Results): *'We then extrapolated the model to estimate the timescales of FLERVs from their S estimates, under an explicit assumption that the TDRP dynamics of mammalian FVs is the same as those infecting their ancient ancestral (perhaps marine) vertebrates'*

Line 327-330 (marked-up 333-336) (Results): *'By assuming that viruses infecting modern-day mammals and their ancient vertebrate ancestors share the same TDRP dynamics,... , the clade of mammalian FVs and lobe-finned fish FLERVs was estimated to be...'*

Line 334-351 (marked-up 340-361) (Results): *'We note that since our tree contains both extant retroviruses (which evolve under pure viral evolutionary rates) and ERVs (which have mixed rates of host and virus evolution) (Fig. 3), some branches in our tree would have mixed rates of virus and host evolution, and because our model was built based on evolutionary dynamics of extant viruses, these mixed rates of ERV evolution could bias our date estimates. However, since we did not impose a molecular clock onto the tree (see **Materials and Methods**), each branch can then vary their length and height independently of time, allowing them to have different rates. Subsequently, those mixed rates of evolution will tend to limit to terminal branches that lead to ERVs. We circumvented this problem of mixed rates of ERV evolution in our date estimation by measuring the node heights from SFVcpz tip down the tree, assuming that internal branches reflect only exogenous viral evolution and do not contain any evolutionary periods in which viruses might have been infectious ERVs.*

As an independent verification, we performed phylogenetic analyses and date estimation by using FLERV Pol ancestral sequences, from which the host evolution has been removed. [We found that the two methods give similar results] Overall, these similarity in the date estimates suggest that we have successfully avoided the problem of mixed rates of ERV evolution in terminal branches.'

We hope that the revised manuscript fully explains all of the concerns raised by the reviewer.

REVIEWERS' COMMENTS:

Reviewer #2 (Remarks to the Author):

The authors have adequately clarified their arguments and addressed our previous concerns.